# Functional proteomic atlas of HIV infection in primary human CD4+ T cells

Adi Naamati[1], James C Williamson[1,2], Edward JD Greenwood[1,2], Sara Marelli[1], Paul J Lehner[2], Nicholas J Matheson[1]*

[1]Department of Medicine, University of Cambridge, Cambridge, United Kingdom; [2]Cambridge Institute for Medical Research, University of Cambridge, Cambridge, United Kingdom

**Abstract** Viruses manipulate host cells to enhance their replication, and the identification of cellular factors targeted by viruses has led to key insights into both viral pathogenesis and cell biology. In this study, we develop an HIV reporter virus (HIV-AFMACS) displaying a streptavidin-binding affinity tag at the surface of infected cells, allowing facile one-step selection with streptavidin-conjugated magnetic beads. We use this system to obtain pure populations of HIV-infected primary human CD4+ T cells for detailed proteomic analysis, and quantitate approximately 9000 proteins across multiple donors on a dynamic background of T cell activation. Amongst 650 HIV-dependent changes (q < 0.05), we describe novel Vif-dependent targets FMR1 and DPH7, and 192 proteins not identified and/or regulated in T cell lines, such as ARID5A and PTPN22. We therefore provide a high-coverage functional proteomic atlas of HIV infection, and a mechanistic account of host factors subverted by the virus in its natural target cell.

DOI: https://doi.org/10.7554/eLife.41431.001

*For correspondence: njm25@cam.ac.uk

**Competing interests:** The authors declare that no competing interests exist.

## Introduction

Remodelling of the host proteome during viral infection may reflect direct effects of viral proteins, secondary effects or cytopathicity accompanying viral replication, or host countermeasures such as the interferon (IFN) response. By defining time-dependent changes in protein levels in infected cells, and correlating temporal profiles of cellular and viral proteins, we have shown that it is possible to differentiate these phenomena, and identify direct cellular targets of human cytomegalovirus (HCMV) and HIV (*Greenwood et al., 2016*; *Matheson et al., 2015*; *Weekes et al., 2014*). To enable time course analysis and minimise confounding effects from uninfected bystander cells, pure populations of synchronously infected cells must be sampled sequentially as they progress through a single round of viral replication. In the case of HIV, we previously satisfied these conditions by spinoculating the highly permissive CEM-T4 lymphoblastoid T cell line (*Foley et al., 1965*; *O'Doherty et al., 2000*; *Popovic et al., 1984*) with Env-deficient NL4-3-ΔEnv-EGFP virus (*Zhang et al., 2004*) at a high multiplicity of infection (MOI) (*Greenwood et al., 2016*).

The utility of cancer cell line models (such as CEM-T4) is, however, limited by the extent to which they retain the characteristics of the primary cells from which they were derived, and cancer-specific and *in vitro* culture-dependent reprogramming are well described (*Gillet et al., 2013*). For example, the HIV accessory proteins Vif, Nef and Vpu are required for viral replication in primary T cells, but not in many T cell lines (*Neil et al., 2008*; *Rosa et al., 2015*; *Sheehy et al., 2002*; *Usami et al., 2015*), and HIV is restricted by type I IFN in primary T cells, but not CEM-derived T cells (*Goujon et al., 2013*). In addition, whilst ensuring a high % infection, dysregulation of the cellular proteome at high MOIs may not be indicative of protein changes when a single transcriptionally active provirus is present per cell.

In this study, we therefore sought to apply our temporal proteomic approach to HIV infection of primary human CD4+ T lymphocytes, the principle cell type infected *in vivo*, at an MOI ≤ 1. To this end, we have developed an HIV reporter virus encoding a cell surface streptavidin-binding affinity tag, allowing antibody-free magnetic cell sorting of infected cells (AFMACS) (*Matheson et al., 2014*) (*Figure 1A*). This system allows rapid, scalable, affinity purification of HIV-infected cells from mixed cultures, bypassing the need for high MOIs or fluorescence-activated cell sorting (FACS). We use this system to generate a detailed atlas of cellular protein dynamics in HIV-infected primary human CD4+ T cells, show how this resource can be used to identify novel cellular proteins regulated by HIV, and assign causality to individual HIV accessory proteins.

## Results

### Design and construction of the HIV-AFMACS reporter virus

AFMACS-based magnetic selection requires the high-affinity 38 amino acid streptavidin-binding peptide (SBP) (*Keefe et al., 2001*) to be displayed at the cell surface by fusion to the N-terminus of the truncated Low-affinity Nerve Growth Factor Receptor (SBP-ΔLNGFR) (*Ruggieri et al., 1997*). Cells expressing this marker may be selected directly with streptavidin-conjugated magnetic beads, washed to remove unbound cells, then released by incubation with the naturally occurring vitamin biotin (*Matheson et al., 2014*). To engineer a single round HIV reporter virus encoding SBP-Δ LNGFR, we considered three settings in the proviral construct: fused to the endogenous Env signal peptide (as a direct replacement for Env); or as an additional cistron, downstream of *nef* and either a P2A peptide or IRES. We used Env-deficient pNL4-3-ΔEnv-EGFP (HIV-1) as a backbone and, since increased size of lentiviral genome is known to reduce packaging efficiency (*Kumar et al., 2001*), tested each approach in constructs from which EGFP was removed and/or the 3' long terminal repeat (LTR) truncated. Further details relating to construct design are described in the Materials and methods and *Supplementary file 1*.

For initial screening, VSVg-pseudotyped viruses were made in HEK-293T cells under standard conditions, and used to spinoculate CEM-T4 T cells (CEM-T4s). Infected cells were identified by expression of EGFP and/or cell surface LNGFR, combined with Nef/Vpu-mediated downregulation of CD4 (*Guy et al., 1987*; *Willey et al., 1992*). Whilst infection is not truly 'productive' (because Env is deleted), Gag alone is sufficient for assembly and release of virions (*Gheysen et al., 1989*), and other structural and non-structural viral proteins are expressed in accordance with full length viral infection (*Greenwood et al., 2016*).

As expected, all viruses tested expressed SBP-ΔLNGFR at the cell surface of infected cells (*Figure 1—figure supplement 1A*), but the larger constructs resulted in lower infectious viral titres (*Figure 1—figure supplement 1A–B*). We therefore selected pNL4-3-ΔEnv-SBP-ΔLNGFR, pNL4-3-Δ ENV-Nef-P2A-SBP-ΔLNGFR and pNL4-3-ΔEnv-Nef-IRES-SBP-ΔLNGFR-Δ3 for further evaluation (*Figure 1—figure supplement 2*). Viruses generated from these constructs expressed high levels of SBP-ΔLNGFR 48 hr post-infection, and depleted CD4 and tetherin to a similar extent. However, only the pNL4-3-ΔENV-Nef-P2A-SBP-ΔLNGFR virus (*Figure 1B*) expressed high levels of LNGFR 24 hr post-infection in both CEM-T4s (*Figure 1—figure supplement 2*) and primary human CD4+ T cells (*Figure 1C*). This is consistent with Nef-P2A-SBP-ΔLNGFR expression from completely spliced transcripts early in HIV infection (*Klotman et al., 1991*), with the P2A peptide ensuring that translation of Nef and SBP-ΔLNGFR follow similar kinetics.

Since analysis of cells at early as well as late time points is essential for the generation of time course data, we focussed on pNL4-3-ΔEnv-Nef-P2A-SBP-ΔLNGFR (now termed HIV-AFMACS). To confirm that HIV-AFMACS virus could be used for cell selection (*Figure 1A*), infected primary T cells were captured by streptavidin-conjugated magnetic beads, released by incubation with excess biotin, then analysed by flow cytometry. Compared with unselected cells (input) or cells released during washing (flow-through), selected cells were markedly enriched for SBP-ΔLNGFR expression and CD4 downregulation (*Figure 1D*). In fact, from mixed populations containing approximately 20–40% infected cells, purities of ≥ 90% were routinely achieved by AFMACS of both CEM-T4s and primary human CD4+ T cells, with ≤ 10% infected cells lost in the flow-through (*Figure 1E*). The full HIV-

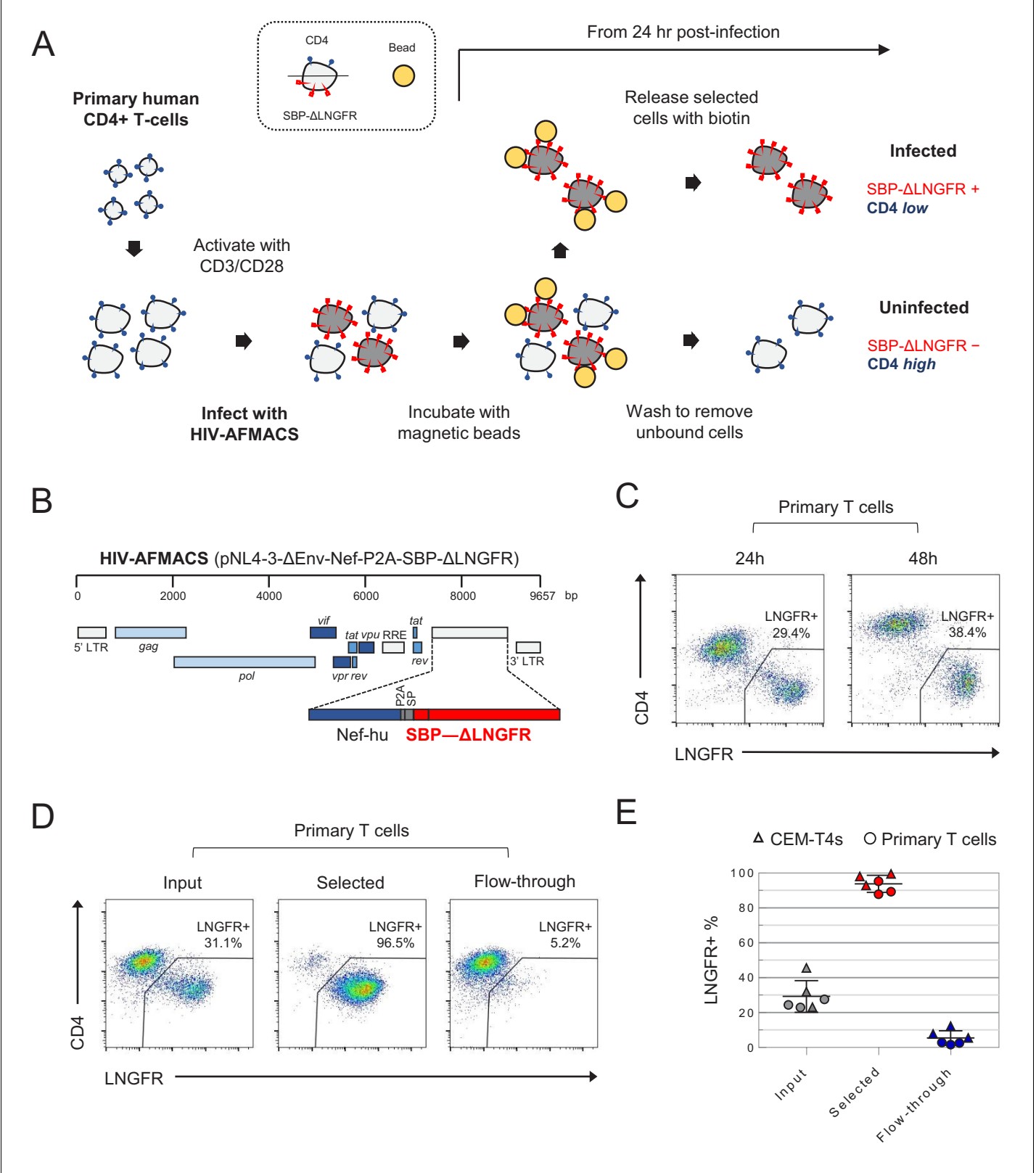

**Figure 1.** Antibody-free magnetic selection of HIV-infected primary T cells. (**A**) Workflow for AFMACS-based magnetic selection of HIV-infected primary T cells. (**B**) Schematic of HIV-AFMACS provirus (pNL4-3-ΔEnv-Nef-P2A-SBP-ΔLNGFR). For simplicity, reading frames are drawn to match the HXB2 HIV-1 reference genome. Length is indicated in base pairs (bp). The complete sequence is available in **Supplementary file 1**. Nef-hu, codon-optimised Nef;
*Figure 1 continued on next page*

Figure 1 continued

RRE, Rev response element; SP, signal peptide. (C) Expression of cell surface SBP-ΔLNGFR and CD4 on primary T cells 24 or 48 hr post-infection with HIV-AFMACS. Cells were stained with anti-LNGFR and anti-CD4 antibodies at the indicated time points and analysed by flow cytometry. (D–E) Magnetic sorting of HIV-infected (red, LNGFR+, CD4 low) and uninfected (blue, LNGFR-, CD4 high) cells. Cells were separated using AFMACS 48 hr post-infection with HIV-AFMACS and analysed as in (C). Representative (D) and summary (E) data from six independent experiments in CEM-T4s (triangles) and primary T cells (circles) are shown, with means and 95% confidence intervals (CIs).

DOI: https://doi.org/10.7554/eLife.41431.002

The following figure supplements are available for figure 1:

**Figure supplement 1.** Initial screen of SBP-ΔLNGFR-expressing HIV viruses.
DOI: https://doi.org/10.7554/eLife.41431.003
**Figure supplement 2.** Time course evaluation of selected SBP-ΔLNGFR-expressing HIV viruses.
DOI: https://doi.org/10.7554/eLife.41431.004

AFMACS sequence is available from GenBank (accession: MK435310) and in *Supplementary file 1*, and the proviral construct will be made available to the community via the National Institutes of Health (NIH) AIDS Reagent Program.

## Time-dependent proteomic remodelling during HIV infection of primary T cells

To gain a comprehensive, unbiased overview of viral and cellular protein dynamics during HIV-infection of its natural target cell, we used the HIV-AFMACS virus to spinoculate activated, primary human CD4+ T cells, sorted infected (SBP-ΔLNGFR positive) and uninfected (SBP-ΔLNGFR negative) cells by AFMACS 24 hr and 48 hr post-infection, and analysed whole cell lysates using tandem mass tag (TMT)-based quantitative proteomics (*Figure 2A–B* and *Figure 2—figure supplement 1A*) (*Greenwood et al., 2016*; *Weekes et al., 2014*). Interpretation of HIV-dependent proteomic remodelling in primary T cells is complicated by concurrent changes in relative protein abundance resulting from T cell activation (*Geiger et al., 2016*). We therefore exploited multiplex TMT labelling to measure parallel protein abundances in resting and activated (uninfected) T cells from the same donor, as well as control (mock-infected) T cells obtained at each time point.

In total, we quantitated 9070 cellular proteins across 10 different conditions. As previously reported (*Geiger et al., 2016*), T cell activation itself caused extensive proteomic remodelling, with relative abundances of 2677/9070 (29%) proteins changing by > 2 fold in activated vs resting cells (*Figure 2—figure supplement 2*). All data from infected and uninfected cells have been made available via ProteomeXchange with identifier PXD012263, and are summarised in an interactive spreadsheet allowing generation of temporal profiles for any quantitated proteins of interest (*Figure 2—source data 1*). For example, the restriction factor tetherin (targeted by HIV-1 Vpu [*Neil et al., 2008*]) is upregulated by T cell activation, then progressively depleted in HIV-infected (red, SBP-ΔLNGFR positive) but not uninfected (blue, SBP-ΔLNGFR negative) cells (*Figure 2C*, left panel). Conversely, the restriction factor SAMHD1 (targeted by some HIV-2/SIV Vpx and Vpr variants, but not HIV-1 [*Hrecka et al., 2011*; *Laguette et al., 2011*; *Lim et al., 2012*]) is depleted by T cell activation, independent of HIV infection (*Figure 2C*, right panel). In these graphical representations, relative protein abundances for each condition are depicted by bars, and ratios of protein abundances in paired experimental/control cells from each condition/time point are depicted by lines (grey, resting/activated; red, SBP-ΔLNGFR positive, infected; blue, SBP-ΔLNGFR negative, uninfected).

Aside from tetherin, levels of many other reported Vpu (CD4, SNAT1) (*Matheson et al., 2015*; *Willey et al., 1992*), Nef (CD4, SERINC5) (*Guy et al., 1987*; *Rosa et al., 2015*; *Usami et al., 2015*), Vif (APOBEC3 and PPP2R5 families) (*Greenwood et al., 2016*; *Sheehy et al., 2002*) and Vpr (UNG, HLTF, ZGPAT, VPRBP, MUS81, EME1, MCM10, TET2) (*Hrecka et al., 2016*; *Lahouassa et al., 2016*; *Lapek et al., 2017*; *Lv et al., 2018*; *Maudet et al., 2013*; *Romani et al., 2015*; *Schröfelbauer et al., 2005*; *Zhou et al., 2016*) substrates were all reduced by HIV infection in primary T cells (*Figure 2D*, and *Figure 2—figure supplement 1B*). Conversely, and consistent with our previous observations in CEM-T4s, APOBEC3B and SERINC1 were not depleted (*Figure 2—figure supplement 1B*) (*Greenwood et al., 2016*; *Matheson et al., 2015*). In the absence of donor haplotyping, polymorphisms at the MHC-I locus make routine proteomic quantification problematic.

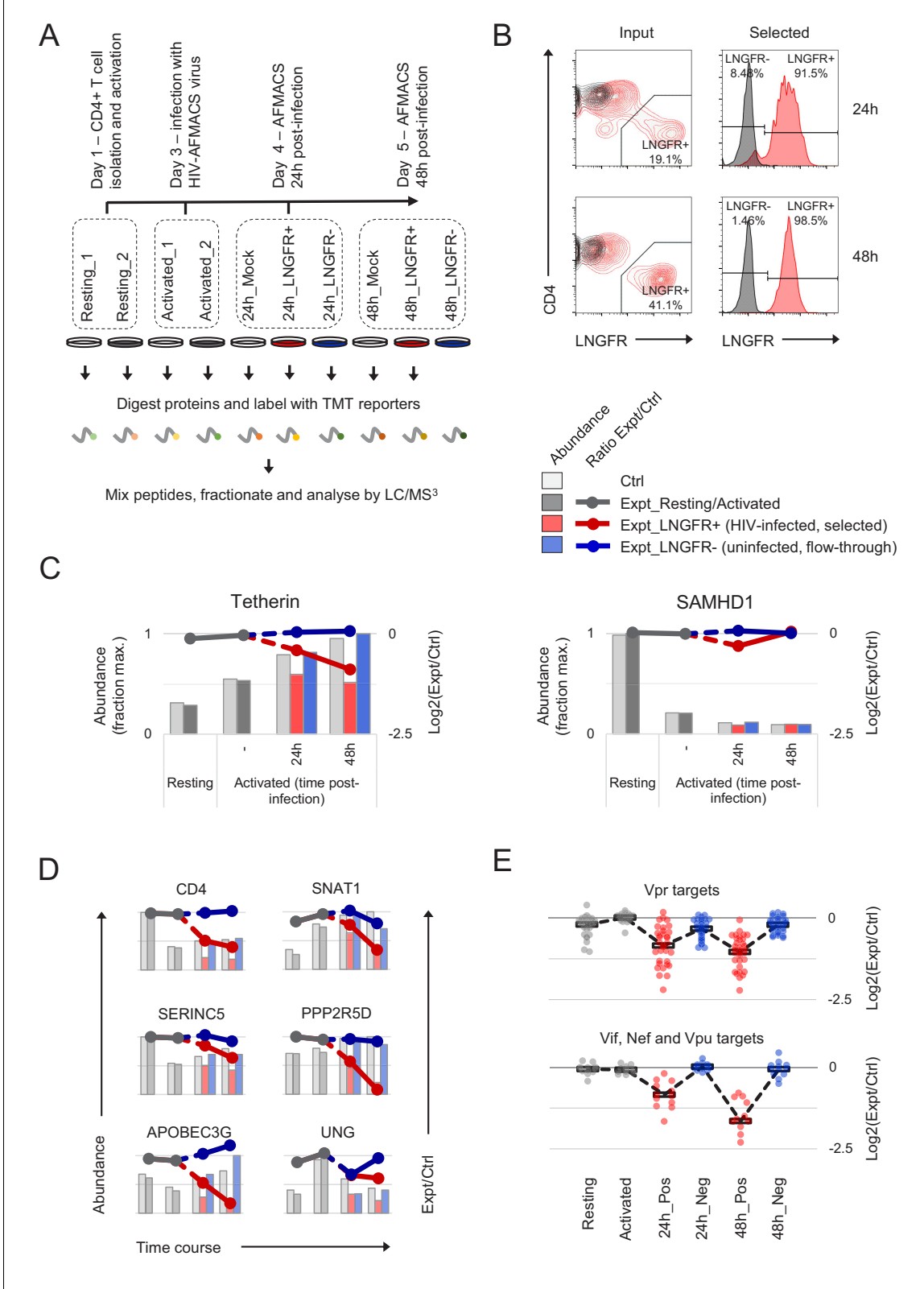

**Figure 2.** Temporal proteomic analysis of HIV infection in primary T cells. (**A**) Overview of time course proteomic experiment. Control (pale grey, resting/activated/mock) and experimental (dark grey, resting/activated; red, LNGFR+, HIV-infected, selected; blue, LNGFR-, uninfected, flow-through) cells are indicated for each condition/time point. (**B**) Magnetic sorting of HIV-infected (red, LNGFR+, selected) cells used for (**A**). Corresponding uninfected (LNGFR-, flow-through) cells are shown in *Figure 2—figure supplement 1A*. Cells were separated using AFMACS at the indicated time

*Figure 2 continued on next page*

*Figure 2 continued*

points post-infection with HIV-AFMACS, stained with anti-LNGFR and anti-CD4 antibodies and analysed by flow cytometry. Mock-infected cells are shown in grey. (C) Expression profiles of illustrative restriction factors regulated by T cell activation and HIV infection (tetherin) or T cell activation alone (SAMHD1) in cells from (A–B). Relative abundances (bars, fraction of maximum) and log2(ratio)s of abundances (lines) in experimental (Expt):control (Ctrl) cells are shown for each condition/time point and coloured as in (A) (summarised in the key). (D) Expression profiles of illustrative accessory protein targets (CD4, Nef/Vpu; SERINC5, Nef; SNAT1, Vpu; APOBEC3G, Vif; PPP2R5D, Vif; UNG, Vpr) in cells from (A–B). Axes, scales and colours are as in (C). Expression profiles of other accessory protein targets are shown in *Figure 2—figure supplement 1B*. (E) Patterns of temporal regulation of Vpr vs other accessory protein (Vif/Nef/Vpu) targets in cells from (A–B). Log2(ratio)s of abundances in experimental (Expt):control (Ctrl) cells are shown for 45 accessory protein targets (as in *Figure 2—figure supplement 3A*). Colours are as in (C), and average profiles (mean, black lozenges/dotted lines) are highlighted for each group of targets.

DOI: https://doi.org/10.7554/eLife.41431.005

The following source data and figure supplements are available for figure 2:

**Source data 1.** Functional proteomic atlas of HIV-infection in primary human CD4+ T cells.
DOI: https://doi.org/10.7554/eLife.41431.006

**Figure supplement 1.** Additional controls for time course proteomic experiment.
DOI: https://doi.org/10.7554/eLife.41431.007

**Figure supplement 2.** Comparison with T cell activation-dependent changes in *Geiger et al. (2016)*.
DOI: https://doi.org/10.7554/eLife.41431.008

**Figure supplement 3.** Temporal clustering of HIV accessory protein targets.
DOI: https://doi.org/10.7554/eLife.41431.009

Nonetheless, our data are consistent with depletion of HLA-A and HLA-B, but not HLA-C (*Figure 2—figure supplement 1B*), as previously reported for Nef/Vpu variants from NL4-3 HIV (*Apps et al., 2016*; *Cohen et al., 1999*; *Schwartz et al., 1996*).

Together with cellular proteins, we identified gene products from seven viral open reading frames (ORFs; *Figure 2—figure supplement 1C*). As expected (*Karn and Stoltzfus, 2012*), viral regulatory and accessory proteins expressed from fully spliced, Rev-independent transcripts (Tat, Rev, Nef-P2A and SBP-ΔLNGFR) were expressed early in infection, peaking at 24 hr. Conversely, viral structural proteins expressed from unspliced, Rev-dependent transcripts (Gag and Gagpol) were expressed late in infection, increasing progressively from 24 to 48 hr. Viral accessory proteins expressed from partially spliced transcripts were either not detected (Vpr and Vpu) or incompletely quantitated (Vif).

## Proteins and pathways regulated by HIV in primary T cells from multiple donors

Inter-individual variability is known to affect gene expression during T cell activation (*Ye et al., 2014*). Accordingly, to identify reproducible HIV targets, we analysed primary human CD4+ T cells from three further donors. In each case, mock-infected cells were compared with HIV-infected cells selected using AFMACS 48 hr post-infection (*Figure 3A–B* and *Figure 3—figure supplement 1A*). Aside from APOBEC3 proteins, we recently discovered the PPP2R5A-E (B56) family of PP2A phosphatase regulatory subunits to be degraded by diverse Vif variants, spanning primate and ruminant lentiviruses (*Greenwood et al., 2016*). To formally document Vif-dependent changes in primary T cells, both wildtype (WT) and Vif-deficient (ΔVif) viruses were therefore included. Whilst some donor-dependent differences were apparent, most sample-sample variability was accounted for by HIV infection (*Figure 3—figure supplement 1B*), and all accessory protein substrates from *Figure 2C–D* and *Figure 2—figure supplement 1B* were significantly depleted by WT HIV (*Figure 3C*, left panel). In total, we quantitated 8789 cellular proteins across nine different conditions, of which 650 were significantly regulated by HIV infection (q < 0.05) and are summarised in an interactive filter table (*Figure 3—source data 1*).

Compared with a previous, similar experiment using CEM-T4s (*Greenwood et al., 2016*), we observed greater variability in protein abundances between replicates (*Figure 3—figure supplement 2A*), but a high degree of correlation in HIV-dependent changes between cell types (*Figure 3—figure supplement 2B*). As well as 'canonical' accessory protein targets, we have recently discovered that most protein-level changes in HIV-infected CEM-T4s may be explained by primary and secondary effects of Vpr, including degradation of at least 34 additional substrates (*Greenwood et al., 2019*). These changes were recapitulated in primary T cells (*Figure 3C*, middle

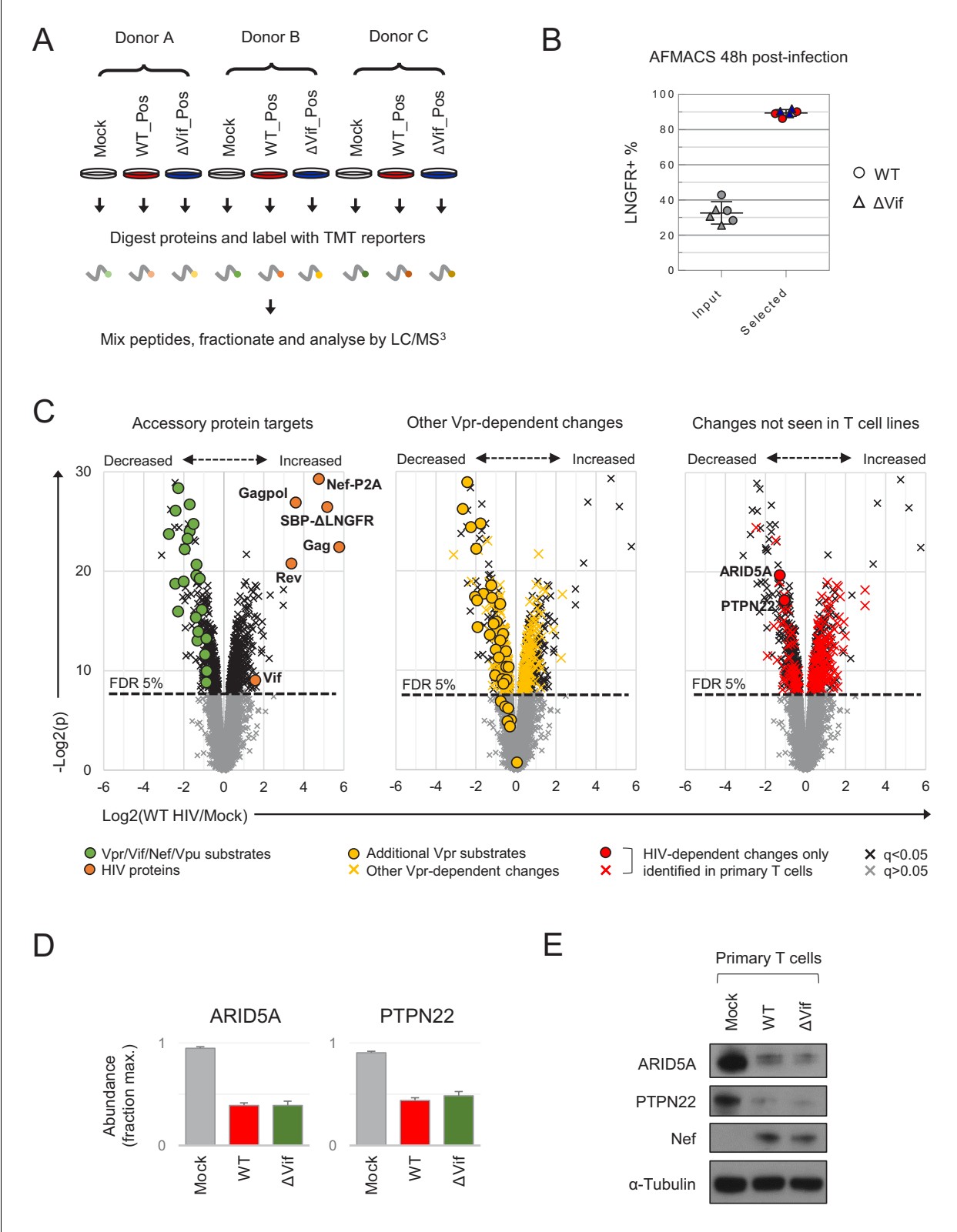

**Figure 3.** Proteins regulated by HIV in primary T cells. (**A**) Overview of single time point proteomic experiment. HIV-infected (LNGFR+) primary T cells were isolated using AFMACS 48 hr post-infection with WT (red) or ΔVif (blue) HIV-AFMACS. (**B**) AFMACS-based enrichment of WT (red circles) and ΔVif (blue triangles) HIV-infected (LNGFR+) cells used for (**A**), with means and 95% CIs. Corresponding cells pre-selection are included for each donor/virus (WT, grey circles; ΔVif, grey triangles). Cells were stained with anti-LNGFR and anti-CD4 antibodies and analysed by flow cytometry, with representative
*Figure 3 continued on next page*

*Figure 3 continued*

data in *Figure 3—figure supplement 1A*. (**C**) Protein abundances in WT HIV-infected vs mock-infected cells from (**A**). Volcano plots show statistical significance (y axis) vs fold change (x axis) for 8789 cellular and six viral proteins quantitated in cells from all three donors (no missing values). Proteins with Benjamini-Hochberg FDR-adjusted p values (q values) < 0.05 or > 0.05 are indicated (FDR threshold of 5%). Proteins highlighted in each plot are summarised in the key. Vpr/Vif/Nef/Vpu substrates (green circles) comprise proteins from *Figure 2C–D* and *Figure 2—figure supplement 1B*, excluding negative controls (SAMHD1, APOBEC3B, SERINC1, HLA-C) and HLA-A/B alleles (different in each donor) but including SMUG1 (not identified in time course proteomic experiment) (*Schröfelbauer et al., 2005*) and both quantitated isoforms of PP2R5C (Q13362 and Q13362-4) and ZGPAT (Q8N5A5 and Q8N5A5-2). Additional Vpr substrates (gold circles) and Vpr-dependent changes (gold crosses) comprise recently described direct and indirect Vpr targets (*Greenwood et al., 2019*). HIV-dependent changes only identified in primary T cells (red circles and crosses) comprise proteins with q < 0.05 either not identified or not concordantly regulated by HIV in CEM-T4s (*Greenwood et al., 2016*) (and exclude known accessory protein-dependent changes). Further details on comparator datasets used in this figure are provided in the Materials and methods. (**D–E**) Abundances of ARID5A and PTPN22 in mock-infected (grey), WT HIV-infected (red) and ΔVif HIV-infected (green) primary T cells from (**A**). Mean abundances (fraction of maximum) with 95% CIs are shown (**D**). As well as proteomic analysis, cells from donor A were lysed in 2% SDS and analysed by immunoblot with anti-ARID5A, anti-PTPN22, anti-Nef and anti-α-tubulin antibodies (**E**). Same lysates as *Figure 5D*.

DOI: https://doi.org/10.7554/eLife.41431.010

The following source data and figure supplements are available for figure 3:

**Source data 1.** Proteins regulated by HIV and/or control lentivectors.
DOI: https://doi.org/10.7554/eLife.41431.011
**Figure supplement 1.** Additional controls for single time point proteomic experiment.
DOI: https://doi.org/10.7554/eLife.41431.012
**Figure supplement 2.** Comparison with HIV-dependent changes in CEM-T4s.
DOI: https://doi.org/10.7554/eLife.41431.013
**Figure supplement 3.** Comparison with HIV-dependent changes in other datasets.
DOI: https://doi.org/10.7554/eLife.41431.014
**Figure supplement 4.** Proteins regulated by transduction with control lentivectors.
DOI: https://doi.org/10.7554/eLife.41431.015
**Figure supplement 5.** Comparison with HIV-dependent changes in *Kuo et al. (2018)*.
DOI: https://doi.org/10.7554/eLife.41431.016

panel), with 33 newly described Vpr substrates quantitated, and 32 decreased in abundance. Several other cell surface proteins reported to be downregulated by Nef and/or Vpu were also depleted, but the magnitude of effect was typically modest, and many were unchanged (*Figure 3—figure supplement 3A*, left panel). Likewise, we did not see evidence of HIV/Vif-dependent transcriptional regulation of RUNX1 target gene products such as T-bet/TBX21 (*Figure 3—figure supplement 3A*, middle panel) (*Kim et al., 2013*). Nonetheless, taken together, known accessory protein-dependent changes, characterised in transformed T cell lines, are able to account for 297/650 (46%) of proteins regulated by HIV in primary T cells (*Figure 3—figure supplement 3B*), including 175/299 (59%) of proteins decreased in abundance.

As with individual proteins, pathways and processes downregulated by HIV infection of primary T cells are dominated by the effects of accessory proteins (*Figure 4A–B*). These include the DNA damage response and cell cycle (Vpr) (*Greenwood et al., 2019*; *He et al., 1995*; *Jowett et al., 1995*; *Laguette et al., 2014*; *Poon et al., 1997*; *Re et al., 1995*; *Rogel et al., 1995*; *Roshal et al., 2003*), cytidine deamination and PP2A activity (Vif) (*Greenwood et al., 2016*; *Harris et al., 2003*; *Sheehy et al., 2002*) and amino acid transport (Vpu/Nef) (*Matheson et al., 2015*). Proteins upregulated by HIV are more diverse, with fewer dominant functional clusters. Nonetheless, we saw marked increases in proteins associated with lipid and sterol metabolism (*Figure 4B–C*). A similar effect has been reported in T cell lines at the transcriptional level, and attributed to the expression of Nef (*Shrivastava et al., 2016*; *van 't Wout et al., 2005*). Similarly, several proteins in these pathways are indirectly regulated by Vpr (*Figure 4C*) (*Greenwood et al., 2019*).

## Identification and characterisation of primary T cell-specific HIV targets

Despite the overall agreement with cell line data, 1252/8789 (14%) cellular proteins quantitated here were not identified in a previous, similar experiment using CEM-T4s (*Greenwood et al., 2016*). Furthermore, having excluded known accessory-protein dependent changes, 192/650 (30%) proteins regulated by HIV in primary T cells were either not detected, or not significantly/concordantly regulated, in CEM-T4s (*Figure 3C*, right panel and *Figure 3—figure supplement 3B*). These proteins

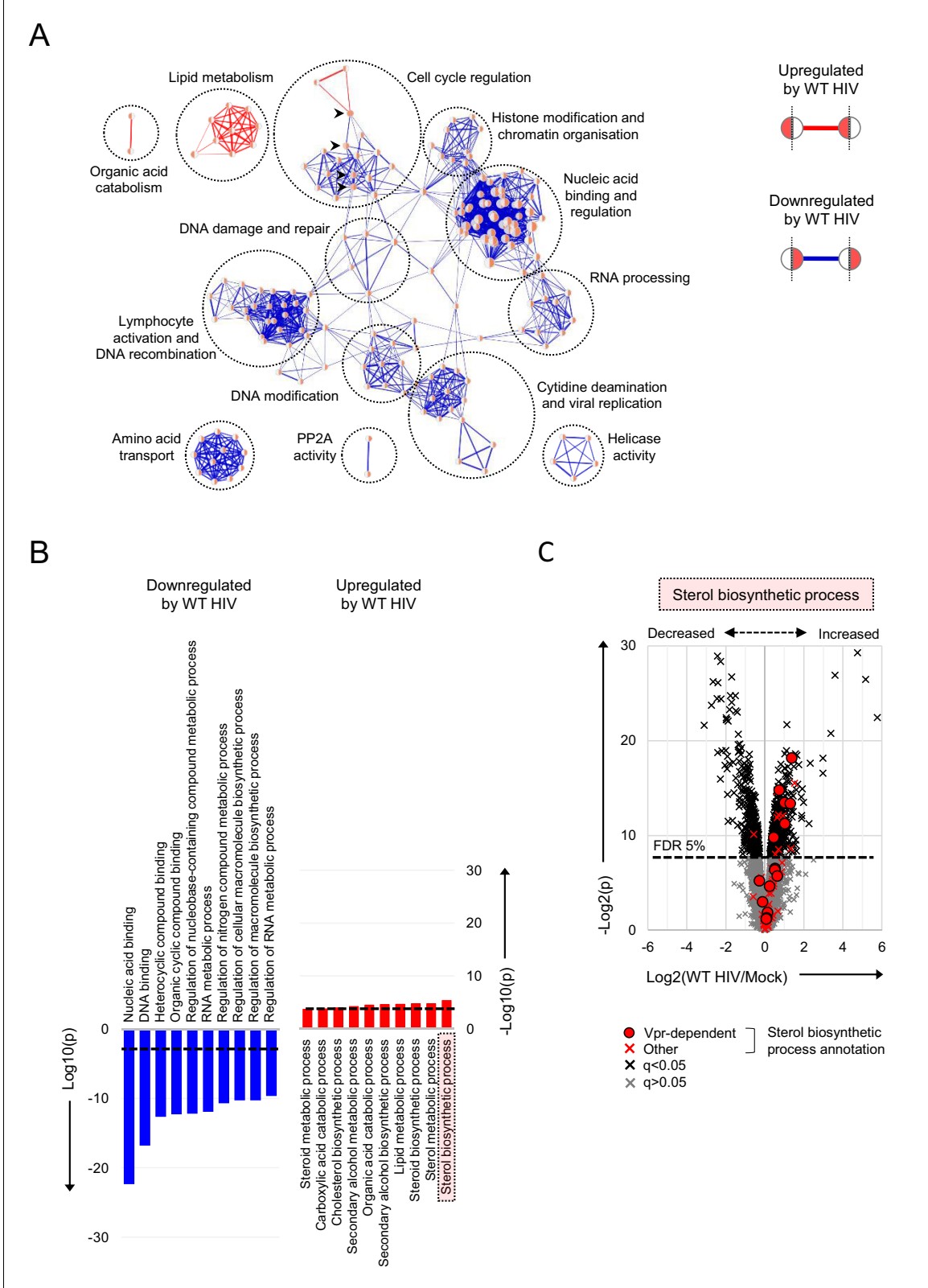

**Figure 4.** Pathways regulated by HIV in primary T cells. (**A–B**) Gene Ontology (GO) functional annotation terms enriched amongst upregulated or downregulated proteins with q < 0.05 in WT HIV-infected vs mock-infected cells from single time point proteomic experiment (**Figure 3A**). In the Enrichment Map (**Merico et al., 2010**) network-based visualisation (**A**), each node represents a GO term, with node size indicating number of annotated proteins, edge thickness representing degree of overlap (red, enriched amongst upregulated proteins; blue, enriched amongst

*Figure 4 continued on next page*

Figure 4 continued

downregulated proteins) and similar GO terms placed close together. Degree of enrichment is mapped to node colour (left side, enriched amongst upregulated proteins; right side, enriched amongst downregulated proteins) as a gradient from white (no enrichment) to red (high enrichment). Highlighted nodes (arrow heads) represent GO terms enriched amongst both upregulated and downregulated proteins. In the bar charts (B), the 10 most enriched GO terms (ranked by p value) amongst upregulated (red) and downregulated (blue) proteins are shown, with an indicative Benjamini-Hochberg FDR threshold of 5% (dashed line). (C) Protein abundances in WT HIV-infected vs mock-infected cells from single time point proteomic experiment (*Figure 3A*), with details for volcano plot as in *Figure 3C*. 57 proteins annotated with the GO term 'sterol biosynthetic process' (GO:0016126) are highlighted in red. Amongst these, 15 proteins are regulated by Vpr in CEM-T4s (circles) (*Greenwood et al., 2019*).

DOI: https://doi.org/10.7554/eLife.41431.017

may represent accessory protein substrates expressed in primary T cells but not T cell lines, or proteins regulated by alternative, cell type-specific mechanisms, such as the interferon response (*Figure 3—figure supplement 3A*, right panel) (*Vermeire et al., 2016*).

We have previously shown that expression of the SBP-ΔLNGFR selectable marker as a transgene does not impact the viability, activation or proliferation of primary T cells (*Matheson et al., 2014*). Nonetheless, some of the novel changes attributed to HIV in this study could theoretically be secondary to exposure to VSVg-pseudotyped viral particles, expression of SBP-ΔLNGFR and/or the AFMACS workflow, or reflect pre-existing proteomic differences in infected (permissive) cells, compared with the mock-infected bulk population. To exclude these possibilities, we repeated the single time point proteomic experiment using primary T cells from three new donors and substituting WT and Vif-deficient HIV-AFMACS for two different control lentivectors expressing SBP-ΔLNGFR either as a single transgene (from the SFFV promoter; pSBP-ΔLNGFR) or in conjunction with HIV-1 Tat (from the HIV-1 LTR; pTat/SBP-ΔLNGFR) (*Figure 3—figure supplement 4A–C*).

As expected, changes in transduced cells were far less extensive than changes induced by HIV (*Figure 3—figure supplement 4D*, top and middle panels; compare with *Figure 3C*). In fact, amongst 8518 cellular proteins quantitated across nine different conditions, only 37/8518 (0.4%) were significantly perturbed by one or both lentivectors (q < 0.05), and are summarised in an interactive filter table (*Figure 3—source data 1*). Interestingly, despite evidence of robust transactivation of the HIV LTR (resulting in high level expression of SBP-ΔLNGFR at the surface of cells transduced with pTat/SBP-ΔLNGFR), no Tat-dependent changes in cellular protein levels were identified (*Figure 3—figure supplement 4C*, lower panels and *Figure 3—figure supplement 4D*, bottom panel). Most importantly, amongst the 650 proteins significantly regulated by HIV, 576 were quantitated in the SBP-ΔLNGFR control experiment, of which only one protein (MYB) was also significantly regulated by the control lentivectors (*Figure 3—figure supplement 4D*, top and middle panels and *Figure 3—source data 1*).

To further validate our proteomic data, we focused on two novel HIV targets with commercially available antibodies: ARID5A and PTPN22. These proteins were readily identified in proteomic datasets from primary T cells (9–14 unique peptides) but not CEM-T4s, and consistently depleted across all donors with a fold change > 2 (*Figure 3D*). As expected, depletion was also seen by immunoblot (*Figure 3E*).

We previously showed that substrates of different HIV accessory proteins could be distinguished by their characteristic patterns of temporal regulation in HIV-infected CEM-T4s (*Greenwood et al., 2016*), and similar clustering was observed in primary T cells (*Figure 2—figure supplement 3A*). Vpr is packaged stoichiometrically in virions (*Cohen et al., 1990*; *Yu et al., 1990*; *Yuan et al., 1990*) and, since the number of fusogenic HIV particles exceeds the infectious MOI by at least several fold (*Thomas et al., 2007*), all cells in our time course experiment were necessarily exposed to incoming Vpr. Accordingly, depletion of known Vpr substrates was near-maximal by 24 hr in infected (red, SBP-ΔLNGFR positive) cells, with partial depletion also seen in uninfected (blue, SBP-ΔLNGFR negative) cells (*Figure 2E*, upper panel). In contrast, since de novo viral protein synthesis is absolutely required, depletion of known Vif, Nef and Vpu substrates increased progressively from 24 to 48 hr, and was only seen in HIV-infected (red, SBP-ΔLNGFR positive) cells (*Figure 2E*, lower panel).

Based on their patterns of temporal regulation, ARID5A and PTPN22 are therefore very likely to represent novel Vpr substrates, specific for primary T cells (*Figure 2—figure supplement 3B*). Consistent with this, another member of the ARID5 subfamily of AT-rich interaction domain (ARID)-

containing proteins, ARID5B, is a widely conserved target of Vpr variants from primate lentivuses (*Greenwood et al., 2019*), and shares a similar temporal profile (*Figure 2—figure supplement 3C*).

## Comprehensive analysis of recognised and novel Vif targets in primary T cells

As predicted, both APOBEC3 and PPP2R5 family proteins were depleted in primary CD4+ T cells infected with WT, but not ΔVif viruses (*Figure 5A–B*). Vif-dependent depletion of PPP2R5A-E causes a marked increase in protein phosphorylation in HIV-infected CEM-T4 T cells, particularly substrates of the aurora kinases (AURKA/B) (*Greenwood et al., 2016*). AURKB activity is enhanced by 'activation loop' auto-phosphorylation at threonine 232 (T232), antagonised by PP2A-B56 (*Meppelink et al., 2015*; *Yasui et al., 2004*). Accordingly, a marked increase in AURKB T232 phosphorylation is seen in CEM-T4s transduced with Vif as a single transgene (*Figure 5C*). We therefore confirmed depletion of PPP2R5D by immunoblot of AFMACS-selected primary T cells and, as a functional correlate, observed increased AURKB phosphorylation (*Figure 5D*).

Besides these known substrates, we also noted differential regulation of several other proteins in primary T cells infected with WT vs ΔVif viruses (*Figure 5A*). Modest changes in PPP2R1A, PPP2R1B and PPP2CA (catalytic/structural subunits of the trimeric PP2A holoenzyme) and PPFIA1 and SGO1 (known PP2A interactors) (*Liu et al., 2014*; *Tang et al., 2006*; *Xu et al., 2009*) are likely to be secondary to destabilisation of PP2A by PPP2R5 depletion, or reflect proximity of the holoenzyme to the Vif-cullin E3 ligase complex (*Figure 5B*). Conversely, DPH7 and FMR1 are not known to interact physically with PP2A, and show more profound and consistent depletion (*Figure 5E*). We therefore suspected these proteins to be novel Vif substrates.

To confirm these findings, we first re-examined our proteomic data from CEM-T4s (*Greenwood et al., 2016*). Unlike ARID5A and PTPN22, DPH7 and FMR1 are expressed in CEM-T4 as well as primary T cells, and only decreased in abundance in HIV-infected cells in the presence of Vif (*Figure 5E*). Next, we confirmed Vif-dependent depletion of both proteins by immunoblot, in cells infected with WT (but not ΔVif viruses) (*Figure 5F*). Finally, we repeated these observations in cells transduced with Vif as a single transgene (*Figure 5G*). Vif is therefore both necessary and sufficient for depletion of DPH7 and FMR1 and, taken together with APOBEC3 and PPP2R5 family proteins/interactors, we can account for all significant Vif-dependent changes in the natural target cell of HIV infection.

## Discussion

Compared with FACS, bead-based magnetic sorting is fast, simple and scalable for simultaneous processing of multiple samples and large cell numbers (*Plouffe et al., 2015*). In conventional, antibody-based immunomagnetic selection, cells remain coated with beads and antibody-antigen complexes, risking alteration of their behaviour or viability through cross-linking of cell-surface receptors or internalisation of ferrous beads (*Bernard et al., 2002*; *Plouffe et al., 2015*; *Stanciu et al., 1996*). Conversely, AFMACS-based selection is antibody free, and selected cells are released from the beads by incubation with biotin, suitable for a full range of downstream applications (*Matheson et al., 2014*). The HIV-AFMACS virus described in this study allows routine isolation of HIV-infected cells subjected to an MOI ≤ 1, avoiding artefacts associated with high MOIs and facilitating experiments in primary cells, where high levels of infection are difficult to achieve in practice.

To demonstrate the utility of this system and provide a resource for the community, we have generated the first high coverage proteomic atlas of HIV-infected primary human CD4+ T cells. As well as identifying HIV-dependent changes in cells from multiple donors, viral regulation may be assessed against a background of endogenous regulation triggered by T cell activation. Unlike T cell lines, primary T cells express a full range of proteins relevant to HIV infection *in vivo*, and are not confounded by the genetic and epigenetic effects of transformation. Furthermore, proteins unique to primary T cells were significantly more likely to be regulated by HIV infection than proteins detected in both primary T cells and CEM-T4s (131/1252 = 10.5% vs 519/7537 = 6.9%; p < 0.0001, two-tailed Fisher's exact test). Our data validate many, but not all, previously reported HIV accessory protein targets. For some Vpu/Nef substrates, such as NTB-A and CCR7, downregulation from the plasma membrane may occur in the absence of protein degradation (*Bolduan et al., 2013*; *Ramirez et al., 2014*; *Shah et al., 2010*). For others, such as ICAM-1/3, accessory protein expression may prevent

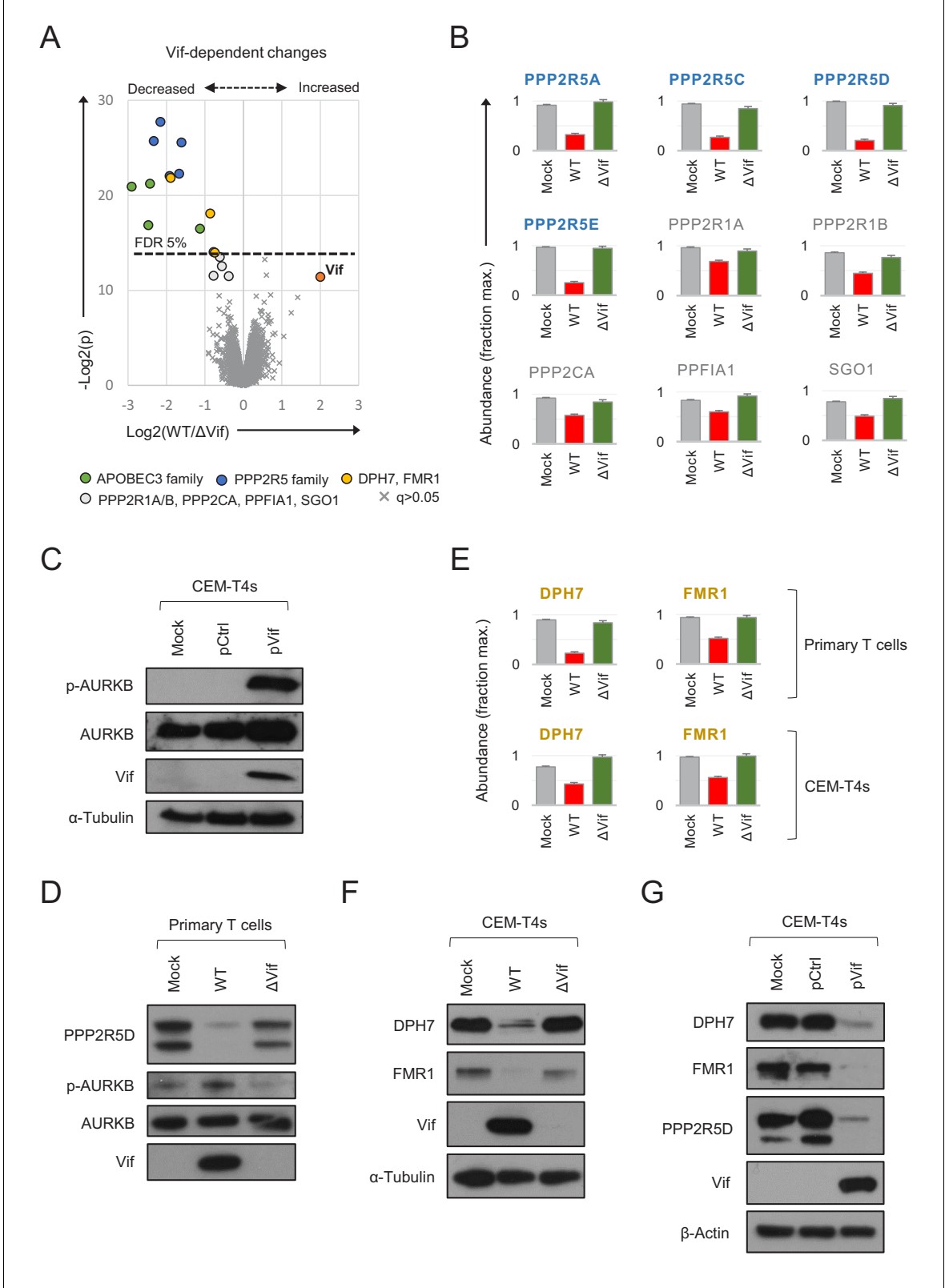

**Figure 5.** Vif-dependent cellular targets in primary T cells. (**A**) Protein abundances in WT HIV-infected vs ΔVif HIV-infected cells from single time point proteomic experiment (*Figure 3A*). Statistical significance (y axis) vs fold change (x axis) is shown for 8795 cellular and viral proteins quantitated in cells from all three donors (no missing values). Proteins with Benjamini-Hochberg FDR-adjusted p values (q values) < 0.05 or > 0.05 are indicated (FDR threshold of 5%). Highlighted groups of differentially regulated proteins are summarised in the key, including two quantitated isoforms of PP2R5C

*Figure 5 continued on next page*

*Figure 5 continued*

(Q13362 and Q13362-4) and FMR1 (Q06787 and Q06787-2). (B) Abundances of Vif-dependent PPP2R5 family and related proteins highlighted in **a**) in mock-infected (grey), WT HIV-infected (red) and ΔVif HIV-infected (green) cells from single time point proteomic experiment (***Figure 3A***). Mean abundances (fraction of maximum) with 95% CIs are shown. Only the canonical isoform of PPP2R5C (Q13362) is shown. (C) Activation of AURKB by Vif. CEM-T4s were mock-transduced or transduced with control (pCtrl) or Vif-expressing (pVif) lentiviruses for 48 hr (62–78% GFP+), lysed in 2% SDS and analysed by immunoblot with anti-phospho-AURK (T232), anti-total AURKB, anti-Vif and anti-α-tubulin antibodies. (D) Vif-dependent activation of AURKB. AFMACS-selected (LNGFR+) HIV-infected cells from ***Figure 3A*** (donor A) were lysed in 2% SDS and analysed by immunoblot with anti-PPP2R5D, anti-phospho-AURK (T232), anti-total AURKB and anti-Vif antibodies. Same lysates as ***Figure 3E***. (E) Abundances of DPH7 and FMR1 in mock-infected (grey), WT HIV-infected (red) and ΔVif HIV-infected (green) primary T cells from single time point proteomic experiment (***Figure 3A***) and CEM-T4s (***Greenwood et al., 2016***). Mean abundances (fraction of maximum) with 95% CIs intervals are shown. Only the canonical isoform of FMR1 (Q06787) is shown. (F) Vif-dependent depletion of DPH7 and FMR1. CEM-T4s were mock-infected or infected with WT or ΔVif HIV-AFMACS viruses for 48 hr (77–82% LNGFR +cells), lysed in 2% SDS, and analysed by immunoblot with anti-DPH7, ant-FMR1, anti-Vif and anti-α-tubulin antibodies. (G) Depletion of DPH7, FMR1 and PPP2R5D by Vif. CEM-T4s were mock-transduced or transduced with control (pCtrl) or Vif-expressing (pVif) lentiviruses for 48 hr (86–88% GFP +cells), lysed in 2% SDS and analysed by immunoblot with anti-phospho-AURK, anti-total AURKB, anti-Vif and anti- α-tubulin (loading control) antibodies.

DOI: https://doi.org/10.7554/eLife.41431.018

upregulation, without reducing levels below baseline (***Sugden et al., 2017***). Alternatively, and particularly where targets have been discovered using model cell line or overexpression systems, regulation may not be quantitatively significant at the protein level in the context and/or natural cell of HIV infection.

Previous temporal proteomic analyses of HIV-infected primary human CD4+ T cells were hampered by extremely limited coverage of the cellular proteome (< 2000 proteins), and did not detect regulation of known or novel HIV accessory protein targets (***Chan et al., 2009***; ***Nemeth et al., 2017***). A more recent study quantitated 7761 proteins in FACS-sorted T cells at a single time point 96 hr post-infection with an R5-tropic, GFP-expressing Nef-deficient virus (***Kuo et al., 2018***). Depletion of several accessory protein targets (including APOBEC3 and PPP2R5 families) was confirmed, and many proteins differentially regulated at 96 hr were also regulated at 48 hr in our study (***Figure 3—figure supplement 5A***). In keeping with the late time point, changes were dominated by pathways involved in cell death and survival, and factors maintaining viability of HIV-infected T cells (such as BIRC5) were enriched. Conversely, the full dataset is not available, effect sizes were generally compressed (***Figure 3—figure supplement 5A***), and 413/650 (64%) of the HIV-dependent changes identified in our study were obscured, including depletion of ARID5A, PTPN22, DPH7 and 19/51 known accessory protein substrates (***Figure 3—figure supplement 5B***).

Compared with other studies, the depth of proteomic coverage reported here not only increases the number of proteins identified, but also reduces the variability in quantitation ('noise'). For example, > 90% of proteins are quantitated using two or more unique peptides. Furthermore, the homogeneous populations of cells analysed (> 90% infected) maximise effect sizes ('signal'), and ensure that proteins exhibiting the most statistically significant differences are also those with the biggest fold changes. Because the 'signal-to-noise ratio' is high, positive controls (known viral targets) behave extremely consistently across our datasets (as shown, for example, in ***Figure 3C*** and ***Figure 2—figure supplement 1B***), and it is possible to make predictions about other cellular proteins falling in the same regions of the volcano plots, and/or exhibiting similar temporal profiles. Our results are therefore useful as a resource (that is, general description of all protein changes), not just as a screen (to identify far outliers).

To enhance viral titers, avoid Env-dependent cytotoxicity, enable synchronous single round infections and bypass co-receptor-dependent targeting of T cell lineages with pre-existing proteomic differences, we used an Env-deficient proviral backbone and pseudotyped viruses with VSVg. Pseudotyping with VSVg redirects HIV viral entry away from the plasma membrane towards an endocytic pathway (***Aiken, 1997***), and may abrogate Env-dependent integrin (***Arthos et al., 2008***) and chemokine co-receptor (***Wu and Yoder, 2009***) signalling early in infection. To limit the impact of these effects, we focussed our analysis on cellular proteins progressively regulated over 48 hr infection. Since only 1/650 HIV-dependent perturbations at this time point was also observed in cells transduced with control lentiviral particles, it is very unlikely that pseudotyping with VSVg per se caused significant artefactual proteomic changes in our datasets (false positives). Nonetheless, it is

possible that the absence of Env-CD4/co-receptor interactions resulted in an underestimate of proteomic changes induced by full length HIV (false negatives), which may vary depending on tropism of the virus (*Wiredja et al., 2018*; *Wojcechowskyj et al., 2013*).

Temporal profiling is particularly well suited to identifying and characterising host factors regulated directly by viral proteins. Because of the need for de novo synthesis of cell surface SBP-Δ LNGFR, it is not possible to perform AFMACS-based selection of HIV-infected primary T cells very early in infection. Nonetheless, even with a first time point 24 hr post-infection, we were able to successfully categorise cellular accessory protein targets according to their patterns of regulation in the time course proteomic experiment. In fact, as we show here, accessory proteins account for much or most of the proteomic remodelling in HIV-infected cells. The abundance of direct accessory protein targets likely explains why proteins and processes/pathways downregulated by HIV in primary T cells correlate so well with changes seen in T cell lines, and are robust to inter-individual variation. In comparison, upregulated proteins concord less well with changes in T cell lines, and functional effects are less homogeneous. This may be because upregulated proteins reflect indirect effects (for example, secondary changes in transcription), which are more likely to be cell-type specific.

Amongst the HIV accessory proteins, Vif was thought until recently to exclusively degrade APOBEC3 family cytidine deaminases. As well as confirming equivalently-potent Vif-dependent depletion of PPP2R5 family phosphatase subunits in primary T cells, our data revealed unexpected Vif-dependent depletion of DPH7 and FMR1. Further work will be required to confirm that these proteins are recruited directly for degradation by the ubiquitin-proteasome system, determine whether (like APOBEC3 and PPP2R5 proteins) they are antagonised by Vif variants from diverse primate and non-primate lentiviral lineages, and identify relevant *in vitro* virological phenotypes associated with target depletion. Nonetheless, FMR1 is already known to reduce HIV virus infectivity when over-expressed in producer cells (*Pan et al., 2009*), and the other novel targets highlighted in this study also impact processes relevant for HIV replication, such as inflammatory cytokine signalling (ARID5A) (*Higa et al., 2018*), T cell activation (PTPN22) (*Hasegawa et al., 2004*) and translational fidelity (DPH7) (*Carette et al., 2009*; *Ortiz et al., 2006*). The diversity of these targets underscores the benefit of an unbiased, systems-level approach to viral infection, and the capacity of the resources presented in this study to reveal unsuspected aspects of the host-virus interaction in the natural target cell of HIV infection.

# Materials and methods

## Key resources table

| Reagent type (species) or resource | Designation | Source or reference | Identifiers | Additional information |
|---|---|---|---|---|
| Cell line | CEM-T4 | NIH AIDS Reagent Program | Cat. #: 117 | |
| Antibody | Mouse monoclonal BV421-conjugated anti-CD4 | BioLegend | Cat. #: 317434 | Flow cytometry |
| Antibody | Mouse monoclonal PE-conjugated anti-CD4 | BD Biosciences | Cat. #: 561843 | Flow cytometry |
| Antibody | Mouse monoclonal PE-conjugated anti-tetherin | BioLegend | Cat. #: 348405 | Flow cytometry |
| Antibody | Mouse monoclonal AF647-conjugated anti-LNGFR | BioLegend | Cat. #: 345114 | Flow cytometry |
| Antibody | Mouse monoclonal FITC-conjugated anti-LNGFR | BioLegend | Cat. #: 345103 | Flow cytometry |

*Continued on next page*

*Continued*

| Reagent type (species) or resource | Designation | Source or reference | Identifiers | Additional information |
|---|---|---|---|---|
| Antibody | Rabbit monoclonal anti-PPP2R5D | Abcam | Cat. #: ab188323 | Immunoblot |
| Antibody | Mouse monoclonal anti-HIV-1 Vif | NIH AIDS Reagent Program | Cat. #: 6459 | Immunoblot |
| Antibody | Mouse monoclonal anti-p24 | Abcam | Cat. #: ab9071 | Immunoblot |
| Antibody | Mouse monoclonal anti-HIV-1 Nef | NIH AIDS Reagent Program | Cat. #: 3689 | Immunoblot |
| Antibody | Rabbit monoclonal anti-PTPN22 (D6D1H) | Cell Signalling Technology | Cat. #: 14693 | Immunoblot |
| Antibody | Mouse monoclonal anti-ARID5A | GeneTex | Cat. #: GTX631940 | Immunoblot |
| Antibody | Rabbit polyclonal anti-FMR1 (FMRP) | Cell Signalling Technology | Cat. #: 4317 | Immunoblot |
| Antibody | Rabbit polyclonal anti-DPH7 | Atlas Antibodies | Cat. #: HPA022911 | Immunoblot |
| Antibody | Mouse monoclonal anti-$\alpha$-tubulin | Cell Signalling Technology | Cat. #: 3873 | Immunoblot |
| Antibody | Mouse monoclonal anti-$\beta$-actin | Sigma | Cat. #: A5316 | Immunoblot |
| Antibody | Rabbit polyclonal anti-total AURKB | Cell Signalling Technology | Cat. #: 3094 | Immunoblot |
| Antibody | Rabbit monoclonal anti-phospho-AURK | Cell Signalling Technology | Cat. #: 2914 | Immunoblot |
| Recombinant DNA reagent | HIV-AFMACS | This paper | GenBank: MK435310 | pNL4-3-$\Delta$Env-Nef-P2A-SBP-$\Delta$LNGFR proviral construct (see Materials and methods) |
| Recombinant DNA reagent | pCtrl | (*Matheson et al., 2014*) | Not applicable | pHRSIN-SE-P2A-SBP-$\Delta$LNGFR-W expression vector |
| Recombinant DNA reagent | pVif | This paper | Not applicable | pHRSIN-SE-P2A-Vif-hu-W expression vector (see Materials and methods) |
| Recombinant DNA reagent | pSBP-$\Delta$LNGFR | This paper | Not applicable | pHRSIN-S-P2A-SBP-$\Delta$LNGFR-W expression vector (see Materials and methods) |
| Recombinant DNA reagent | pTat/SBP-$\Delta$LNGFR | This paper | Not applicable | pLTR-Tat-P2A-SBP-$\Delta$LNGFR expression vector (see Materials and methods) |
| Commercial assay or kit | Dynabeads Untouched Human CD4 T Cells kit | Invitrogen | Cat. #: 11346D | |
| Commercial assay or kit | Dynabeads Human T-Activator CD3/CD28 | Gibco | Cat. #: 11132D | |
| Commercial assay or kit | Dynabeads Biotin Binder | Invitrogen | Cat. #: 11047 | |
| Commercial assay or kit | iST-NHS Sample Preparation Kit | PreOmics | Cat. #: P.O.00030 | |

*Continued on next page*

*Continued*

| Reagent type (species) or resource | Designation | Source or reference | Identifiers | Additional information |
|---|---|---|---|---|
| Commercial assay or kit | S-Trap micro MS Sample Preparation Kit | Protifi | Cat. #: C02-micro | |
| Commercial assay or kit | TMT10plex Isobaric Label Reagent Set | Thermo Scientific | Cat. #: 90110 | |
| Chemical compound, drug | Lympholyte-H | Cedarlane Laboratories | Cat. #: CL5020 | |
| Chemical compound, drug | IL-2 | PeproTech | Cat. #: 200–02 | Recombinant human IL-2 |
| Chemical compound, drug | Lenti-X Concentrator | Clontech | Cat. #: 631232 | |
| Software, algorithm | Proteome Discoverer 2.1 | Thermo Scientific | RRID: SCR_014477 | |
| Software, algorithm | DAVID 6.8 | (*Huang et al., 2009a*; *Huang et al., 2009b*) | RRID: SCR_001881 | https://david.ncifcrf.gov/ |
| Software, algorithm | Cytoscape 3.6.1 | (*Shannon et al., 2003*) | RRID: SCR_003032 | http://cytoscape.org/ |
| Software, algorithm | Enrichment Map 3.1.0 Cystoscape plugin | (*Merico et al., 2010*) | RRID: SCR_016052 | http://baderlab.org/Software/EnrichmentMap |
| Software, algorithm | Cluster 3.0 | (*de Hoon et al., 2004*) | RRID: SCR_013505 | http://bonsai.hgc.jp/~mdehoon/software/cluster/software.htm |
| Software, algorithm | Java TreeView 1.1.6r4 | (*Saldanha, 2004*) | RRID:SCR_013503 | http://jtreeview.sourceforge.net |

## General cell culture

CEM-T4 T cells (CEM-T4s) (*Foley et al., 1965*) were obtained directly (< 1 year) from the AIDS Reagent Program, Division of AIDS, NIAD, NIH: Dr J.P. Jacobs and cultured in RPMI supplemented with 10% FCS, 100units/ml penicillin and 0.1 mg/ml streptomycin at 37°C in 5% $CO_2$. HEK-293T cells were obtained from Lehner laboratory stocks (authenticated by STR profiling [*Menzies et al., 2018*; *Miles et al., 2017*]) and cultured in DMEM supplemented with 10% FCS, 100units/ml penicillin and 0.1 mg/ml streptomycin at 37°C in 5% $CO_2$. All cells were confirmed to be mycoplasma negative (Lonza MycoAlert).

## Primary cell isolation and culture

Primary human CD4+ T cells were isolated from peripheral blood by density gradient centrifugation over Lympholyte-H (Cedarlane Laboratories) and negative selection using the Dynabeads Untouched Human CD4 T Cells kit (Invitrogen) according to the manufacturer's instructions. Purity was assessed by flow cytometry for CD3 and CD4 and routinely found to be ≥ 95%. Cells were activated using Dynabeads Human T-Activator CD3/CD28 beads (Gibco) according to the manufacturer's instructions and cultured in RPMI supplemented with 10% FCS, 30 U/ml recombinant human IL-2 (PeproTech), 100units/ml penicillin and 0.1 mg/ml streptomycin at 37°C in 5% $CO_2$.

### Ethics statement

Ethical permission for this study was granted by the University of Cambridge Human Biology Research Ethics Committee (HBREC.2017.20). Written informed consent was obtained from all volunteers prior to providing blood samples.

## HIV-1 molecular clones

pNL4-3-ΔEnv-EGFP (*Zhang et al., 2004*) was obtained from the AIDS Reagent Program, Division of AIDS, NIAD, NIH: Drs Haili Zhang, Yan Zhou, and Robert Siliciano and the complete proviral sequence verified by Sanger sequencing (Source BioScience). Derived from the HIV-1 molecular clone pNL4-3, it encodes EGFP in the *env* ORF), resulting in a large, critical *env* deletion and expression of a truncated Env-EGFP fusion protein retained in the endoplasmic reticulum (ER) by a 3' KDEL ER-retention signal.

The SBP-ΔLNGFR selection marker comprises the high-affinity 38 amino acid SBP fused to the N-terminus of a truncated (non-functional) member of the Tumour Necrosis Factor Receptor super-family (LNGFR) (*Matheson et al., 2014*). As a type I transmembrane glycoprotein, expression at the cell surface requires a 5' signal peptide.

To replace EGFP with SBP-ΔLNGFR (generating pNL4-3-ΔEnv-SBP-ΔLNGFR) a synthetic gene fragment (gBlock; Integrated DNA Technologies, IDT) was incorporated into pNL4-3-ΔEnv-EGFP by Gibson assembly between SalI/BsaBI sites (gBlock #1; *Supplementary file 1*). In this construct, SBP-ΔLNGFR is fused with the endogenous Env signal peptide.

To express SBP-ΔLNGFR downstream of *nef* and a 'self-cleaving' *Porcine teschovirus-1* 2A (P2A) peptide (generating pNL4-3-ΔEnv-EGFP-Nef-P2A-SBP-ΔLNGFR) a gBlock (IDT) was incorporated into pNL4-3-ΔEnv-EGFP by Gibson assembly between HpaI/XhoI sites (gBlock #2; *Supplementary file 1*). In this construct, SBP-ΔLNGFR is co-translated with codon-optimised Nef (Nef-hu) and includes an exogenous murine immunoglobulin (Ig) signal peptide. SBP-ΔLNGFR was located downstream (rather than upstream) of Nef-hu to avoid disruption of Nef myristoylation by addition of a 5' proline residue following P2A 'cleavage'.

To express SBP-ΔLNGFR downstream of *nef* and an *Encephalomyocarditis virus* (EMCV) internal ribosome entry site (IRES; generating pNL4-3-ΔEnv-EGFP-Nef-IRES-SBP-ΔLNGFR) a gBlock (IDT) was incorporated into pNL4-3-ΔEnv-EGFP by Gibson assembly between HpaI/XhoI sites (gBlock #3; *Supplementary file 1*). In this construct, SBP-ΔLNGFR is translated independently of Nef-hu and includes an exogenous murine Ig signal peptide. A widely-used replication-competent HIV EGFP reporter virus was previously generated using a similar approach (*Schindler et al., 2006*; *Schindler et al., 2003*).

In all constructs, Nef or Nef-hu expression is mediated by the WT HIV LTR promoter and naturally occurring splice sites. In constructs with a P2A peptide or IRES, the use of codon-optimised Nef-hu minimises homology with the U3 region of the 3' LTR (overlapped by the endogenous *nef* sequence) and reduces the risk of recombination.

To remove EGFP from constructs with a P2A peptide or IRES, a gBlock (IDT) was incorporated by Gibson assembly between SalI/BsaBI sites (gBlock #4; *Supplementary file 1*). To avoid generating a truncated protein product fused to the Env signal peptide, the *env* start codon and other potential out of frame start codons in the *vpu* ORF were disrupted with point mutations, whilst maintaining the Vpu protein sequence. Redundant *env* sequence was minimised, without disrupting the Rev response element (RRE).

To truncate the U3 region of the 3' LTR in constructs with a P2A peptide or IRES (with or without EGFP), a gBlock (IDT) was incorporated by Gibson assembly between XhoI/NaeI sites (gBlock #5; *Supplementary file 1*). Previous studies have shown that, in the presence of an intact *nef* ORF, the overlap between *nef* and the U3 region is dispensable for HIV gene expression and replication (*Münch et al., 2005*).

To generate a Vif-deficient HIV-AFMACS molecular clone (pNL4-3-ΔVif-ΔEnv-Nef-P2A-SBP-ΔLNGFR), a restriction fragment encoding a stop codon early in the Vif ORF (after the final in-frame start codon) was subcloned from pNL4-3-ΔVif-ΔEnv-EGFP (*Greenwood et al., 2016*) into pNL4-3-ΔEnv-Nef-P2A-SBP-ΔLNGFR (HIV-AFMACS) between AgeI/PflMI sites.

Where appropriate, additional unique restriction sites were included to facilitate future cloning. All sequences were verified by Sanger sequencing (Source BioScience). The complete HIV-AFMACS sequence is available in *Supplementary file 1*.

## Lentivectors for transgene expression

pHRSIN-SE-P2A-SBP-ΔLNGFR-W (referred to as pCtrl in this paper, in which EGFP and SBP-ΔLNGFR expression are mediated by the spleen focus-forming virus (SFFV) promoter and coupled by a P2A peptide) has been previously described (*Matheson et al., 2014*).

For over-expression of SBP-ΔLNGFR as a single transgene, overlapping DNA oligomers (Sigma) encoding a short peptide linker were incorporated into pCtrl in place of EGFP by restriction cloning between BamHI/NotI sites to generate pHRSIN-S-P2A-SBP-ΔLNGFR-W (referred to as pSBP-ΔLNGFR in this paper).

For over-expression of HIV-1 Tat and SBP-ΔLNGFR from the HIV-1 LTR promoter, P2A-SBP-ΔLNGFR was PCR-amplified from HIV-AFMACS and incorporated into pLTR-Tat-IRES-GFP (pEV731, a kind gift from Eric Verdin [*Jordan et al., 2001*]) by Gibson assembly with a bridging gBlock (IDT) between ClaI/XhoI sites. In this construct (pLTR-Tat-P2A-SBP-ΔLNGFR, referred to as pTat/SBP-ΔLNGFR in this paper), Tat and SBP-ΔLNGFR expression are coupled by a P2A peptide, replacing Tat-IRES-GFP in the original lentivector.

For over-expression of codon optimised NL4-3 Vif (Vif-hu), a gBlock (IDT) was incorporated into pCtrl in place of SBP-ΔLNGFR by Gibson assembly between KpnI/XhoI sites to generate pHRSIN-SE-P2A-Vif-hu-W (referred to as pVif in this paper, in which EGFP and Vif-hu expression are coupled by a P2A peptide).

## Viral stocks

VSVg-pseudotyped NL4-3-ΔEnv-based viral stocks were generated by co-transfection of HEK-293 T cells with pNL4-3-ΔEnv molecular clones and pMD.G at a ratio of 9:1 (µg) DNA and a DNA:FuGENE 6 ratio of 1 µg:6 µl. Media was changed the next day and viral supernatants harvested and filtered (0.45 µm) at 48 hr prior to concentration with Lenti-X Concentrator (Clontech) and storage at −80°C.

VSVg-pseudotyped lentivector stocks were generated by co-transfection of 293Ts with lentivector, p8.91 and pMD.G at a ratio of 2:1:1 (µg) DNA and a DNA:FuGENE 6 ratio of 1 ug:3 µl. Viral supernatants were harvested, filtered, concentrated and stored as per NL4-3-ΔEnv-based viral stocks.

All viruses and lentivectors were titered by infection/transduction of known numbers of relevant target cells with known volumes of viral stocks under standard experimental conditions, followed by flow cytometry for SBP-ΔLNGFR or EGFP plus/minus CD4 at 48 hr to identify the fraction of infected cells (f) containing at least one transcriptionally active provirus (SBP-ΔLNGFR or EGFP positive plus/minus CD4 low). The number of infectious/transducing units present was then calculated by assuming a Poisson distribution (where $f = 1-e^{-MOI}$). Typically, a dilution series of each viral stock was tested, and titer determined by linear regression of $-\ln(1-f)$ on volume of virus.

## T cell infections

CEM-T4s were infected/transduced by spinoculation at 800 g for 2 hr in a non-refrigerated benchtop centrifuge in complete media supplemented with 10 mM HEPES. Primary human CD4+ T cells were infected/transduced using the same protocol 48 hr after activation with CD3/CD28 Dynabeads.

Unlike CEM-T4s, permissivity of infected primary T cells varies between donors/experiments, and the maximum fraction of infected cells in viral dilution series is often around 50% for single round infections, even at high MOI. In practice, we therefore aimed to use sufficient infectious/transducing units to achieve approximately 30% infection, corresponding to a 'nominal' MOI $\leq$ 0.5 (assuming a Poisson distribution). This ensured that, even if only 50% of cells were permissive, the 'effective' MOI would still be $\leq$ 1.

## Antibody-Free magnetic cell sorting (AFMACS)

AFMACS-based selection of CEM-T4 or primary human CD4+ T cells using the streptavidin-binding SBP-ΔLNGFR affinity tag was carried out essentially as previously described (*Matheson et al., 2014*). For primary T cells, CD3/CD28 Dynabeads were first removed using a DynaMag-2 magnet (Invitrogen). 24 or 48 hr post-infection, washed cells were resuspended in incubation buffer (IB; Hank's balanced salt solution, 2% dialysed FCS, 1x RPMI Amino Acids Solution (Sigma), 2 mM L-glutamine, 2 mM EDTA and 10 mM HEPES) at 10e7 cells/ml and incubated with Dynabeads Biotin Binder (Invitrogen) at a bead-to-total cell ratio of 4:1 for 30 min at 4°C. Bead-bound cells expressing SBP-ΔLNGFR

were selected using a DynaMag-2 (Invitrogen), washed to remove uninfected cells, then released from the beads by incubation in complete RPMI with 2 mM biotin for 15 min at room temperature (RT). Enrichment was routinely assessed by flow cytometry pre- and post-selection.

## Proteomic analysis

### Sample preparation

For TMT-based whole cell proteomic analysis of primary human CD4+ T cells, resting or activated cells were washed with ice-cold PBS with Ca/Mg pH 7.4 (Sigma) and frozen at −80°C. Samples were lysed, reduced, alkylated, digested and labelled with TMT reagents (Thermo Scientific) using either iST-NHS (PreOmics GmbH; time course and single time point experiments) or S-Trap (Protifi; SBP-Δ LNGFR control experiment) sample preparation kits, according to the manufacturers' instructions. Typically, 5e6 resting or 1e6 activated cells were used for each condition.

### Off-line high pH reversed-phase (HpRP) peptide fractionation

HpRP fractionation was conducted on an Ultimate 3000 UHPLC system (Thermo Scientific) equipped with a 2.1 mm ×15 cm, 1.7 µm Acquity BEH C18 column (Waters, UK). Solvent A was 3% ACN, solvent B was 100% ACN, and solvent C was 200 mM ammonium formate (pH 10). Throughout the analysis C was kept at a constant 10%. The flow rate was 400 µL/min and UV was monitored at 280 nm. Samples were loaded in 90% A for 10 min before a gradient elution of 0–10% B over 10 min (curve 3), 10–34% B over 21 min (curve 5), 34–50% B over 5 min (curve 5) followed by a 10 min wash with 90% B. 15 s (100 µL) fractions were collected throughout the run. Peptide-containing fractions were orthogonally recombined into 24 fractions (e.g. fractions 1, 25, 49, 73 and 97) and dried in a vacuum centrifuge. Fractions were stored at −80°C prior to analysis.

### Mass spectrometry

Data were acquired on an Orbitrap Fusion mass spectrometer (Thermo Scientific) coupled to an Ultimate 3000 RSLC nano UHPLC (Thermo Scientific). HpRP fractions were resuspended in 20 µl 5% DMSO 0.5% TFA and 10 uL injected. Fractions were loaded at 10 µl/min for 5 min on to an Acclaim PepMap C18 cartridge trap column (300 um ×5 mm, 5 um particle size) in 0.1% TFA. Solvent A was 0.1% FA and solvent B was ACN/0.1% FA. After loading, a linear gradient of 3–32% B over 3 hr was used for sample separation over a column of the same stationary phase (75 µm × 50 cm, 2 µm particle size) before washing at 90% B and re-equilibration.

An SPS/MS3 acquisition was used for all samples and was run as follows. MS1: Quadrupole isolation, 120'000 resolution, 5e5 AGC target, 50 ms maximum injection time, ions injected for all parallelisable time. MS2: Quadrupole isolation at an isolation width of m/z 0.7, CID fragmentation (NCE 35) with the ion trap scanning out in rapid mode from m/z 120, 8e3 AGC target, 70 ms maximum injection time, ions accumulated for all parallelisable time. In synchronous precursor selection mode the top 10 MS2 ions were selected for HCD fragmentation (65NCE) and scanned out in the orbitrap at 50'000 resolution with an AGC target of 2e4 and a maximum accumulation time of 120 ms, ions were not accumulated for all parallelisable time. The entire MS/MS/MS cycle had a target time of 3 s. Dynamic exclusion was set to ±10 ppm for 90 s, MS2 fragmentation was trigged on precursor ions 5e3 counts and above.

### Data processing and analysis

Spectra were searched by Mascot within Proteome Discoverer 2.1 in two rounds. The first search was against the UniProt Human reference proteome (26/09/17), the HIV-AFMACS proteome and compendium of common contaminants (GPM). The second search took all unmatched spectra from the first search and searched against the human trEMBL database (Uniprot, 26/09/17). For time course and single time point experiments, the following search parameters were used. MS1 Tol: 10 ppm. MS2 Tol: 0.6 Da. Fixed Mods: Ist-alkylation (+113.084064 Da) (C) and TMT (N-term, K). Var Mods: Oxidation (M). Enzyme: Trypsin (/P). For the SBP-ΔLNGFR control experiment, Carbamidomethyl (C) modification was used in place of Ist-Alkylation. MS3 spectra were used for reporter ion-based quantitation with a most confident centroid tolerance of 20 ppm. Peptide spectrum match (PSM) false discovery rate (FDR) was calculated using Mascot percolator and was controlled at 0.01% for 'high' confidence PSMs and 0.05% for 'medium' confidence PSMs. Normalisation was

automated and based on total s/n in each channel. Proteins/peptides satisfying at least a 'medium' FDR confidence were taken forth to statistical analysis in R. This consisted of a moderated T-test (Limma) with Benjamini-Hochberg correction for multiple hypotheses to provide a q value for each comparison (*Schwämmle et al., 2013*). Further data manipulation and general statistical analysis (including principal component analysis) was conducted using Excel, XLSTAT and GraphPad Prism 7.

All mass spectrometry proteomics data from this study have been deposited to the ProteomeX-change Consortium via the PRIDE (*Vizcaíno et al., 2016*) partner repository with the dataset identifier PXD012263 and 10.6019/PXD012263 (accessible at http://proteomecentral.proteomexchange.org).

For functional analysis of proteins significantly downregulated or upregulated by WT HIV (q < 0.05) in the single time point proteomic experiment (*Figure 3A*), enrichment of Gene Ontology (GO) Biological Process (GOTERM_BP_FAT) and Molecular Function (GOTERM_MF_FAT) terms against a background of all proteins quantitated was determined using the Database for Annotation, Visualization and Integrated Discovery (DAVID) 6.8 (accessed on 22/7/2018 at https://david.ncifcrf.gov/) with default settings (*Huang et al., 2009a*; *Huang et al., 2009b*). Human proteins annotated to GO:0016126 (sterol biosynthetic process) were retrieved from AmiGO 2 (accessed on 27/7/2018 at http://amigo.geneontology.org/amigo) (*Carbon et al., 2009*). To account for redundancy between annotations, enriched GO terms were visualised using the Enrichment Map 3.1.0 plugin (*Merico et al., 2010*) for Cytoscape 3.6.1. (downloaded from http://cytoscape.org/) (*Shannon et al., 2003*) with default settings (q value cut-off of 0.1) and sparse-intermediate connectivity. Clusters were manually labelled to highlight the prevalent biological functions amongst each set of related annotations.

For clustering according to profiles of temporal expression, known accessory protein substrates from *Figure 2C–D* and *Figure 2—figure supplement 1B* and additional Vpr substrates shown in *Figure 3C* were analysed using Cluster 3.0 (downloaded from http://bonsai.hgc.jp/~mdehoon/software/cluster/software.htm) (*de Hoon et al., 2004*) and visualised using Java TreeView 1.1.6r4 (downloaded from http://jtreeview.sourceforge.net) (*Saldanha, 2004*). Only proteins significantly downregulated by WT HIV (q < 0.05) in the single time point proteomic experiment (*Figure 3A*) were included. Where more than one isoform was quantitated, only the canonical isoform was used (PPP2R5C, Q13362; ZGPAT, Q8N5A5; NUSAP1, Q9BXS6). Data from the time course proteomic experiment (*Figure 2A*) were expressed as log2(ratio)s of abundances in experimental (Expt):control (Ctrl) cells for each condition/time point, and range-scaled to highlight patterns of temporal expression relative to the biological response range (minimum-maximum) for each protein. Agglomerative hierarchical clustering was performed using uncentered Pearson correlation and centroid linkage (*Eisen et al., 1998*).

## Comparison with CEM-T4 T cells

To compare results between primary and transformed T cells at a similar depth of proteomic coverage, we re-analysed TMT-labelled peptide eluates from a previous study (*Greenwood et al., 2016*) conducted in CEM-T4s spinoculated in triplicate with VSVg-pseudotyped NL4-3-ΔEnv-EGFP WT or ΔVif viruses at an MOI of 1.5. This extended analysis consisted of reinjection of HpRP fractions on longer (3 hr) gradients using a higher performance (75 as opposed to 50 cm) analytical column and the MS parameters employed in this study. In total, the new CEM-T4 dataset covered 8065 proteins, comparable with the datasets from primary T cells described here.

## Comparisons with other published datasets

A previous study quantitated 7816 proteins at multiple time points following *in vitro* activation of naïve (CD45RA+ CCR7+) primary human CD4+ human T cells with plate-bound anti-CD23 and anti-CD28 antibodies (*Geiger et al., 2016*). For comparison with this study, a filtered list of 5907 proteins quantitated in at least two samples from both resting cells and cells activated for 48 hr was used.

We have recently characterised 34 new Vpr substrates, together with further, extensive Vpr-dependent changes (downregulated and upregulated proteins) in HIV-infected CEM-T4s (*Greenwood et al., 2019*). For comparison with this study, a list of 1388 proteins concordantly regulated by Vpr (q < 0.05) in the context of both viral infection and Vpr-bearing virus-like particles was compiled from the published datasets.

RUNX1 target genes were previously found to be regulated by Vif at a transcriptional level because of competition for CBFβ binding (*Kim et al., 2013*). For comparison with this study, the reported list of 155 genes with RUNX1-associated regulatory domains exhibiting Vif-dependent differential gene expression in Jurkat T cells after 4 or 6 hr of PMA and PHA treatment was used.

Curated lists of ISGs have been previously described (*Schoggins et al., 2014*; *Schoggins et al., 2011*). For comparison with this study, a list of 377 ISGs was compiled from these studies.

A recent study quantitated 7761 proteins in FACS-sorted T cells at a single time point 96 hr post-infection with an R5-tropic, GFP-expressing Nef-deficient virus (*Kuo et al., 2018*). The comparator is GFP negative rather than mock-infected cells (equivalent to SBP-ΔLNGFR negative cells in this study), and the full dataset is not available. For comparison with this study, the published list of 1551 differentially expressed proteins (q < 0.05) was therefore used.

## Flow cytometry

For primary T cells, CD3/CD28 Dynabeads were first removed using a DynaMag-2 magnet (Invitrogen). Typically 2e5 washed cells were incubated for 15 min in 100 μL PBS with the indicated fluorochrome-conjugated antibody. All steps were performed on ice or at 4°C and stained cells were fixed in PBS/1% paraformaldehyde.

## Immunoblotting

Cells were lysed in PBS/2% SDS supplemented with Halt Protease Inhibitor Cocktail (Thermo Scientific) and Halt Phosphatase Inhibitor Cocktail (Thermo Scientific) for 10 min at RT. Benzonase (Sigma) was included to reduce lysate viscosity. Post-nuclear supernatants were heated in Laemelli Loading Buffer for 25 min at 50°C, separated by SDS-PAGE and transferred to Immobilon-P membrane (Millipore). Membranes were blocked in PBS/5% non-fat dried milk (Marvel)/0.2% Tween and probed with the indicated primary antibody overnight at 4°C. Reactive bands were visualised using HRP-conjugated secondary antibodies and SuperSignal West Pico or Dura chemiluminescent substrates (Thermo Scientific). Typically 10–20 μg total protein was loaded per lane (Pierce BCA Protein Assay kit).

## Antibodies

Antibodies for immunoblot and flow cytometry are detailed in the Key resources table. The following antibodies were obtained from the AIDS Reagent Program, Division of AIDS, NIAID, NIH: mouse monoclonal anti-HIV-1 Vif (*Simon et al., 1995*) from Dr MH Malim, and mouse monoclonal anti-HIV-1 Nef (*Chang et al., 1998*) from Dr JA Hoxie.

# Acknowledgements

This work was supported by the MRC (CSF MR/P008801/1 to NJM), NHSBT (WPA15-02 to NJM), the Wellcome Trust (PRF 210688/Z/18/Z to PJL), the NIHR Cambridge BRC, and a Wellcome Trust Strategic Award to CIMR. The authors thank Dr Reiner Schulte and the CIMR Flow Cytometry Core Facility team, and members of the Lehner laboratory for critical discussion.

# Additional information

### Funding

| Funder | Grant reference number | Author |
|---|---|---|
| Medical Research Council | MR/P008801/1 | Nicholas J Matheson |
| NHS Blood and Transplant | WPA15-02 | Nicholas J Matheson |
| Wellcome | 210688/Z/18/Z | Paul J Lehner |

The funders had no role in study design, data collection and interpretation, or the decision to submit the work for publication.

## Author contributions
Adi Naamati, Conceptualization, Formal analysis, Validation, Investigation, Visualization, Methodology, Writing—original draft, Writing—review and editing; James C Williamson, Data curation, Formal analysis, Supervision, Investigation, Methodology, Writing—review and editing; Edward JD Greenwood, Conceptualization, Investigation, Writing—review and editing; Sara Marelli, Conceptualization, Validation, Investigation, Writing—review and editing; Paul J Lehner, Conceptualization, Resources, Supervision, Funding acquisition, Project administration, Writing—review and editing; Nicholas J Matheson, Conceptualization, Resources, Data curation, Formal analysis, Supervision, Funding acquisition, Visualization, Methodology, Writing—original draft, Project administration, Writing—review and editing

## Author ORCIDs
Edward JD Greenwood (iD) http://orcid.org/0000-0002-5224-0263
Paul J Lehner (iD) http://orcid.org/0000-0001-9383-1054
Nicholas J Matheson (iD) https://orcid.org/0000-0002-3318-1851

## Ethics
Human subjects: Ethical permission for this study was granted by the University of Cambridge Human Biology Research Ethics Committee (HBREC.2017.20). Written informed consent was obtained from all volunteers prior to providing blood samples.

## Decision letter and Author response
Decision letter https://doi.org/10.7554/eLife.41431.025
Author response https://doi.org/10.7554/eLife.41431.026

# Additional files

## Supplementary files
• Supplementary file 1. gBlock and HIV-AFMACS sequences.
DOI: https://doi.org/10.7554/eLife.41431.019

• Transparent reporting form
DOI: https://doi.org/10.7554/eLife.41431.020

## Data availability
All data generated or analysed during this study are included in the manuscript and supporting files. Source data files have been provided for Figures 2 and 3. All mass spectrometry proteomics data have been deposited to the ProteomeXchange Consortium via the PRIDE partner repository with the dataset identifier PXD012263 and 10.6019/PXD012263 (accessible at http://proteomecentral.proteomexchange.org).

The following dataset was generated:

| Author(s) | Year | Dataset title | Dataset URL | Database and Identifier |
|---|---|---|---|---|
| Naamati A, Williamson JC, Greenwood EJD, Marelli S | 2018 | Functional proteomic atlas of HIV infection in primary human CD4+ T cells | http://proteomecentral.proteomexchange.org/cgi/GetDataset?ID=PXD012263 | ProteomeXchange Consortium , PXD012263 |

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
