## [Decision Letter]

Thank you for submitting your article "Functional proteomic atlas of HIV infection in primary human CD4^+^ T cells" for consideration by *eLife*. Your article has been reviewed Wenhui Li as the Senior Editor, a Reviewing Editor, and three reviewers. The following individual involved in review of your submission has agreed to reveal his identity: Gregory J Towers (Reviewer #3).

The reviewers have discussed the reviews with one another and the Reviewing Editor has drafted this decision to help you prepare a revised submission.

Summary:

While there was discussion among the reviewers regarding the biological significance of your findings, all 3 were impressed with the magnitude of your data set using primary CD4^+^ T cells. They felt that it was technically innovative, that the mass spec data had depth and stringency, and that you have provided an easy-to-access resource for the community. We have therefore agreed in principle to go forward, potentially publishing your manuscript as a Tool and Resources paper rather than a Research Advance (https://submit.elifesciences.org/html/eLife_author_instructions.html#types).

"Tools and Resources articles do not have to report major new biological insights or mechanisms, but it must be clear that they will enable such advances to take place. Specifically, these contributions will be assessed in terms of their potential to facilitate experiments that address problems that to date have been very challenging or even intractable." (https://lens.elifesciences.org/07083/)

Please go through the specific comments of the reviewers below and send us a revised manuscript with itemized responses to each comment, and formatting with a eye towards publishing your manuscript as a Tool and Resource paper.

Reviewer #1:

Naamati and coworkers use a quantitative temporal proteomics approach to survey protein-level changes in primary CD4^+^ cells infected with HIV-1 plus/minus various accessory genes. This is the largest data set in using primary CD4^+^ T cells to-date and over half of the 600+ changes were dependent on Vpr and Vif/Nef/Vpu. This is a technically sound study and a potential resource for the community studying HIV/host interactions.

However, my main concern is novelty and overlap with prior and parallel work. For instance, an *eLife* paper a couple of years ago focused on Vif by the same group, a PLOS One paper focused on infected cell purification by the same group, and a sister paper (Greenwood, 2018) that describes the Vpr downregulated proteome. The authors indicate 40-60% Venn overlap with proteomic studies in CEM cells, and focus on novel cellular factors (ARID5A, PTPN22, DPH7, and FMR1) but the functional significance of these interactions is unclear.

Reviewer #2:

Naamati et al., present a strategy for isolating HIV-infected primary cells for mass spec analysis based on infecting cells with modified virus bearing a genetically-encoded surface protein with a strong affinity to streptavidin. This virus ("HIV-AFMACS") allows for one-step enrichment of infected cells using magnetic beads. The authors go on to show the utility of this strategy by measuring changes in cellular protein levels upon infection of primary cells with this construct at 24 and 48 hours post-infection, and by comparing changes in protein expression using WT and ΔVif viruses. The authors complete a thorough comparison of the resulting data in the context of previous studies in the literature. They also validate the changes in expression of two proteins that were specific to primary cells (as opposed to CEM-T4s) and two proteins that are potential new Vif substrates using immunoblot.

The paper is clearly written and presented, and its strengths include, in particular, (1) the technical innovation, (2) the emphasis on primary cells, and (3) the depth and stringency of the mass spec data obtained. Especially notable is the interactive spreadsheet that provides an easy-to-access and expansive resource.

Weaknesses of the study are, at a fundamental level, relatively minor. However, they are worth pondering in the context of the anticipated target audience.

First, it can be argued that there is not much new biological insight gleaned here, especially compared to the depth of the stories in these authors' other recent (Vif- and Vpu-focused) and concurrent (Vpr-focused) proteomics studies using CEMs. The major advance seems to be the technical achievement and comparison of primary T cells to cell lines, and while this is both compelling and innovative, there is rather limited validation of new or interesting hits and no characterization of why they (in particular the highlighted factors DPH7 and FMR1) are affected by HIV or the roles they might play in viral or cellular biology.

Second, it is unfortunate that ΔEnv viruses were chosen for the analysis considering the large impact Env-CD4/CoR binding can have on primary T cell responses, likely highly relevant to accurately modeling acute infection dynamics (e.g., see Wojcechowskyj et al., 2013). To their credit, the authors discuss the VSV-G issue (albeit briefly), saying that because of VSV-G they biased their focus to be on late stage accessory gene contributions (Discussion section). However, it seems reasonable that their viruses could have been pseudotyped with wild-type Envelope proteins instead of VSV-G. Indeed, again in the context of acute infection, it would have been even more compelling to compare R5 vs. X4 or TF to non-TF glycoproteins. Further discussion of this issue would be warranted.

Third, the authors did not include an evaluation of whether the expression of SBP-LNGFR and its display on the cell surface causes any changes in expression of cellular proteins, as an important negative control. Unless I missed it, the only cells that are expressing this protein are also infected with HIV, therefore it is not possible to fully distinguish which proteins are changing expression due to HIV infection or SBP-LNGFR expression alone. While the authors do discuss how their technique is superior to antibodies that might cross-link surface receptors, they should also explain the limitations of AFMACS-selection more thoroughly and whether or not the potential effects of LNGFR overexpression and bead binding can truly be disregarded.

Reviewer #3:

In this study Naamati and colleagues measure the proteomic changes experienced by T cells after HIV infection. The study is compelling, appropriately controlled and the results are of significant interest to the community. I have some minor suggestions to improve clarity.

1) The authors make a lot of the MOI being low. Do they know that this is true? Permissivity is typically different between donors/experiments but typically one cannot infect all the cells, especially for primary T cells, even by spinoculation with VSV-G. In the case that only 40% of the cells are permissive an MOI of 50 would only infect 40% of the cells. In order to determine actual MOI one has to titrate the virus back to make sure that eg halving the dose halves the number of infected cells. In the experiments presented, do the authors know that reducing dose reduces infectivity or is their MOI of 0.5 really just an indication that only half of the cells are permissive. Did they test whether lower dose give predictably lower numbers of infected cells? This is important because it also speaks to whether the only difference between the infected and uninfected cells is chance, ie there is not enough virus to infect all the cells, or are the uninfected cells non permissive. In this case, gene expression differences between uninfected and infected cells may be as much to do with the cells being different as it is to do with viral gene expression. I think the experiments are OK and this has been taken into account but a more explicit discussion of this point and whether diluting the virus reduces infectivity as expected is important.

2) A key goal of this study is to provide a resource for examining which proteins are manipulated by HIV infection. With this in mind, could the authors annotate the data in Figure 3C with more detail. Each outlier circle could be numbered and a table of gene names provided. I appreciate that the authors are presenting all of their data and are not hiding anything. But I feel the outliers are what people really want to know about and these could be labeled here. In fact, any opportunity to label the volcano plots would enormously improve the accessibility and thus the likelihood that the field will chase these hits up mechanistically.

3) It’s not immediately obvious what "pos" and "neg" mean in Figure 2A. For clarity, label "HIV+" and "HIV-" instead and change the text in subsection “Time-dependent proteomic remodelling during HIV infection of primary T cells” to "whole cell lystates from both HIV positive and HIV negative populations using.….

4) Provide key for blue and red lines in Figure 2C-E on Figure and in legend.

5) Subsection “Design and construction of the HIV-AFMACS reporter virus” submit the sequence of the construct to GenBank and provide accession number which is more useful than having the seq in a Figure.

6) Subsection “Proteins and pathways regulated by HIV in primary T cells from multiple donors”, its useful to label the x for the constructs that didn't work. Knowing what didn't work is useful, particularly for those that were completely dead vs a bit defective. Figure 1—figure supplement 1B.

7) Can the authors comment on the impact of T cell activation of HIV permissivity. It’s not totally clear why T cells have to be activated to make them permissive for HIV infection. SAMHD1 has a lot to do with it and the authors make this point. In these experiments T cell receptor is crosslinked with anti-CD3, anti-CD28 dyna beads. This presumably doesn't happen *in vivo*, yet T cells are permissive. I imagine the authors have thought about this and I would be interested to hear their thoughts, perhaps in the Discussion section. Is there value in their data comparing unactivated and activated cells to consider what's driving permissivity? How do their data compare with any published literature on *in vivo* activated T cells, indeed is there any data on this? I'm interested to hear what they think and whether they think their data can be used to illuminate the changes that occur on TCR cross linking that drive permissivity. Would a figure considering this point be valuable?

---

## [Author Response]

Summary:While there was discussion among the reviewers regarding the biological significance of your findings, all 3 were impressed with the magnitude of your data set using primary CD4^+^ T cells. They felt that it was technically innovative, that the mass spec data had depth and stringency, and that you have provided an easy-to-access resource for the community. We have therefore agreed in principle to go forward, potentially publishing your manuscript as a Tool and Resources paper rather than a Research Advance (https://submit.elifesciences.org/html/eLife_author_instructions.html#types)."Tools and Resources articles do not have to report major new biological insights or mechanisms, but it must be clear that they will enable such advances to take place. Specifically, these contributions will be assessed in terms of their potential to facilitate experiments that address problems that to date have been very challenging or even intractable." (https://lens.elifesciences.org/07083/)Please go through the specific comments of the reviewers below and send us a revised manuscript with itemized responses to each comment, and formatting with an eye towards publishing your manuscript as a Tool and Resource paper.

Thank you for the kind comments and helpful suggestion. We agree that it would be appropriate to consider our manuscript as a Tool and Resources paper. As suggested, we have therefore responded to specific comments with this in mind, and adjusted the abstract to fit this format.

In particular, we have performed a further large-scale proteomic experiment to validate the HIV-AFMACS approach and formally exclude artefacts (summarised in Figure 3—figure supplement 4 and described in the responses to reviewers #2 and #3), and added an interactive filter table summarising HIV-dependent changes to facilitate data mining by other groups (Figure 3—source data 1, described in the responses to reviewer #3).

We have also uploaded our proteomic data to the ProteomeXchange Consortium, deposited the HIV-AFMACS sequence to GenBank, and will distribute the construct to the community via the NIH AIDS Reagent Program. Relevant dataset/sequence identifiers are included in the revised manuscript, and critical reagents are now summarised in a Key resources table at the start of the Materials and methods section.

Reviewer #1:Naamati and coworkers use a quantitative temporal proteomics approach to survey protein-level changes in primary CD4^+^ cells infected with HIV-1 plus/minus various accessory genes. This is the largest data set in using primary CD4^+^ T cells to-date and over half of the 600+ changes were dependent on Vpr and Vif/Nef/Vpu. This is a technically sound study and a potential resource for the community studying HIV/host interactions.However, my main concern is novelty and overlap with prior and parallel work. For instance, an eLife paper a couple of years ago focused on Vif by the same group, a PLOS One paper focused on infected cell purification by the same group, and a sister paper (Greenwood, 2018) that describes the Vpr downregulated proteome. The authors indicate 40-60% Venn overlap with proteomic studies in CEM cells, and focus on novel cellular factors (ARID5A, PTPN22, DPH7, and FMR1) but the functional significance of these interactions is unclear.

We agree in general terms with the reviewer’s analysis and, as stated above, are happy for our manuscript to be considered as a Tools and Resources paper. In that regard, and together with the proteomic atlas, the HIV-AFMACS virus certainly does facilitate experiments in primary human CD4^+^ T cells which have previously been very technically challenging (as demonstrated). As mentioned by the reviewer, we originally developed AFMACS to isolate cells transfected or transduced with expression vectors (Matheson et al., 2014), before appreciating the potential of the system for the study of HIV infection in its natural target cell.

Whilst supported by our work in CEM-T4s, we here identify 192 novel HIV-dependent changes in primary human CD4^+^ T cells (30% of all dysregulated proteins), including the specific examples listed by the reviewer. These proteins are now all summarised for easy reference in Figure 3—source data 1 (described in the responses to reviewer #3). It is of course reassuring (and important *per se*) that there is a strong correlation between primary T cell and cell line data. Conversely, we found proteins unique to primary T cells significantly more likely to be perturbed by HIV infection, and several previously reported direct and indirect HIV targets are not regulated at all in the primary T cell system, at least at the total protein level (illustrated in Figure 3—figure supplement 3A).

Reviewer #2:Naamati et al., present a strategy for isolating HIV-infected primary cells for mass spec analysis based on infecting cells with modified virus bearing a genetically-encoded surface protein with a strong affinity to streptavidin. This virus ("HIV-AFMACS") allows for one-step enrichment of infected cells using magnetic beads. The authors go on to show the utility of this strategy by measuring changes in cellular protein levels upon infection of primary cells with this construct at 24 and 48 hours post-infection, and by comparing changes in protein expression using WT and ΔVif viruses. The authors complete a thorough comparison of the resulting data in the context of previous studies in the literature. They also validate the changes in expression of two proteins that were specific to primary cells (as opposed to CEM-T4s) and two proteins that are potential new Vif substrates using immunoblot.The paper is clearly written and presented, and its strengths include, in particular, (1) the technical innovation, (2) the emphasis on primary cells, and (3) the depth and stringency of the mass spec data obtained. Especially notable is the interactive spreadsheet that provides an easy-to-access and expansive resource.Weaknesses of the study are, at a fundamental level, relatively minor. However, they are worth pondering in the context of the anticipated target audience.First, it can be argued that there is not much new biological insight gleaned here, especially compared to the depth of the stories in these authors' other recent (Vif- and Vpu-focused) and concurrent (Vpr-focused) proteomics studies using CEMs. The major advance seems to be the technical achievement and comparison of primary T cells to cell lines, and while this is both compelling and innovative, there is rather limited validation of new or interesting hits and no characterization of why they (in particular the highlighted factors DPH7 and FMR1) are affected by HIV or the roles they might play in viral or cellular biology.

As above, we agree in general terms with the reviewer’s analysis, and are happy for our manuscript to be considered as a Tool and Resources paper. Whilst we have not focused on functional follow-up of individual “hits” here, we do think that biological insight can also be gained at a systems level. On our previous experience with this sort of study (alluded to by the reviewer), the follow-up often becomes the main story, distracting from the resource-value of novel reagents, techniques and datasets. By taking a different approach in this manuscript, we ultimately hope to facilitate more detailed functional follow-up by our lab and others.

Second, it is unfortunate that ΔEnv viruses were chosen for the analysis considering the large impact Env-CD4/CoR binding can have on primary T cell responses, likely highly relevant to accurately modeling acute infection dynamics (e.g., see Wojcechowskyj et al., 2013). To their credit, the authors discuss the VSV-G issue (albeit briefly), saying that because of VSV-G they biased their focus to be on late stage accessory gene contributions (Discussion section). However, it seems reasonable that their viruses could have been pseudotyped with wild-type Envelope proteins instead of VSV-G. Indeed, again in the context of acute infection, it would have been even more compelling to compare R5 vs. X4 or TF to non-TF glycoproteins. Further discussion of this issue would be warranted.

Thank you for this comment, we agree that it merits more detailed discussion than originally provided. First, and most importantly, we considered it important to exclude the possibility of “false positive” proteomic changes, related to exposure to VSVg-pseudotyped viral particles, rather than HIV infection *per se* (analogous to a Type I error). We therefore performed a further large scale proteomic experiment to address this directly. The results are shown in new Figure 3—figure supplement 4 (proteins regulated by transduction with control lentivectors) and Figure 3—source data 1 (proteins regulated by HIV and/or control lentivectors) and described in the text (subsection “Identification and characterisation of primary T cell-specific HIV targets”):

“We have previously shown that expression of the SBP-ΔLNGFR selectable marker as a transgene does not impact the viability, activation or proliferation of primary T cells (Matheson et al., 2014). Nonetheless, some of the novel changes attributed to HIV in this study could theoretically be secondary to exposure to VSVg-pseudotyped viral particles, expression of SBP-ΔLNGFR and/or the AFMACS workflow, or reflect pre-existing proteomic differences in infected (permissive) cells, compared with the mock-infected bulk population. To exclude these possibilities, we repeated the single time point proteomic experiment using primary T cells from 3 new donors and substituting WT and Vif-deficient HIV-AFMACS for two different control lentivectors expressing SBP-ΔLNGFR either as a single transgene (from the SFFV promoter; pSBP-ΔLNGFR) or in conjunction with HIV-1 Tat (from the HIV-1 LTR; pTat/SBP-ΔLNGFR) (Figure 3—figure supplement 4A-C).

As expected, changes in transduced cells were far less extensive than changes induced by HIV (Figure 3—figure supplement 4D, top and middle panels; compare with Figure 3C). In fact, amongst 8,518 cellular proteins quantitated across 9 different conditions, only 37/8,518 (0.4%) were significantly perturbed by one or both lentivectors (q<0.05), and are summarised in an interactive filter table (Figure 3—source data 1). Interestingly, despite evidence of robust transactivation of the HIV LTR (resulting in high level expression of SBP-ΔLNGFR at the surface of cells transduced with pTat/SBP-ΔLNGFR), no Tat-dependent changes in cellular protein levels were identified (Figure 3—figure supplement 4C, lower panels and Figure 3—figure supplement 4D, middle and bottom panels). Most importantly, amongst the 650 proteins significantly regulated by HIV, 576 were quantitated in the SBP-ΔLNGFR control experiment, of which only 1 protein (MYB) was also significantly regulated by the control lentivectors (Figure 3—figure supplement 4D, top and middle panels and Figure 3—source data 1).”

Second, we agree that the use of VSVg-pseudotyped particles introduces the potential for “false negative” proteomic changes, because of the absence of potential HIV-Env dependent changes (analogous to a Type II error). To highlight this and the first point, we have therefore also extended the Discussion section as follows:

“To enhance viral titers, avoid Env-dependent cytotoxicity, enable synchronous single round infections and bypass co-receptor-dependent targeting of T cell lineages with pre-existing proteomic differences, we used an Env-deficient proviral backbone and pseudotyped viruses with VSVg. Pseudotyping with VSVg redirects HIV viral entry away from the plasma membrane towards an endocytic pathway (Aiken, 1997), and may abrogate Env-dependent integrin (Arthos et al., 2008) and chemokine co-receptor (Wu and Yoder, 2009) signalling early in infection. To limit the impact of these effects, we focussed our analysis on cellular proteins progressively regulated over 48 hrs infection. Since only 1/650 HIV-dependent perturbations at this time point was also observed in cells transduced with control lentiviral particles, it is very unlikely that pseudotyping with VSVg *per se* caused significant artefactual proteomic changes in our datasets (false positives). Nonetheless, it is possible that the absence of Env-CD4/co-receptor interactions resulted in an underestimate of proteomic changes induced by full length HIV (false negatives), which may vary depending on tropism of the virus (Wiredja et al., 2018; Wojcechowskyj et al., 2013).”

Of note, Wojcechowskyj et al., 2013 (for example) quantitated early phosphoproteomic changes in resting primary T cells exposed to HIV virions without the need for cell selection (because all cells present were exposed to virions, even if not actively infected). Direct changes arising from the Env-CD4/co-receptor interactions could likely be assessed using a similar approach i.e. analysis of bulk populations. In fact, AFMACS is not possible at very early time points post-infection, because of the requirement for de novo synthesis of SBP-ΔLNGFR.

Third, the authors did not include an evaluation of whether the expression of SBP-LNGFR and its display on the cell surface causes any changes in expression of cellular proteins, as an important negative control. Unless I missed it, the only cells that are expressing this protein are also infected with HIV, therefore it is not possible to fully distinguish which proteins are changing expression due to HIV infection or SBP-LNGFR expression alone. While the authors do discuss how their technique is superior to antibodies that might cross-link surface receptors, they should also explain the limitations of AFMACS-selection more thoroughly and whether or not the potential effects of LNGFR overexpression and bead binding can truly be disregarded.

Again, we agree that this is an important point, which we have therefore now addressed with the SBP-ΔLNGFR control proteomic experiment outlined in Figure 3—figure supplement 4 (proteins regulated by transduction with control lentivectors) and Figure 3—source data 1 and described above. In addition, and as above, we note that the utility of AFMACS-based selection is limited at very early time points, because of the need for *de novo* synthesis of SBP-dLNGFR. Accordingly, we have added the following to the Discussion section:

“Temporal profiling is particularly well suited to identifying and characterising host factors regulated directly by viral proteins. Because of the need for de novo synthesis of cell surface SBP-ΔLNGFR, it is not possible to perform AFMACS-based selection of HIV-infected primary T cells very early in infection. Nonetheless, even with a first time point 24 hrs post-infection, we were able to successfully categorise cellular accessory protein targets according to their patterns of regulation in the time course proteomic experiment.”

Reviewer #3:In this study Naamati and colleagues measure the proteomic changes experienced by T cells after HIV infection. The study is compelling, appropriately controlled and the results are of significant interest to the community. I have some minor suggestions to improve clarity.1) The authors make a lot of the MOI being low. Do they know that this is true? Permissivity is typically different between donors/experiments but typically one cannot infect all the cells, especially for primary T cells, even by spinoculation with VSV-G. In the case that only 40% of the cells are permissive an MOI of 50 would only infect 40% of the cells. In order to determine actual MOI one has to titrate the virus back to make sure that eg halving the dose halves the number of infected cells. In the experiments presented, do the authors know that reducing dose reduces infectivity or is their MOI of 0.5 really just an indication that only half of the cells are permissive. Did they test whether lower dose give predictably lower numbers of infected cells? This is important because it also speaks to whether the only difference between the infected and uninfected cells is chance, ie there is not enough virus to infect all the cells, or are the uninfected cells non permissive. In this case, gene expression differences between uninfected and infected cells may be as much to do with the cells being different as it is to do with viral gene expression. I think the experiments are OK and this has been taken into account but a more explicit discussion of this point and whether diluting the virus reduces infectivity as expected is important.

Again, we agree that this is an important point, and it is something we were careful to account for when planning the experiments but did not describe in sufficient detail. We always titer our viruses in the relevant cell type, using a dilution series, to ensure that the number of infected cells is titratable around the range of infectious/transducing units used. We have now added a much more detailed description to the Materials and methods section, which specifically addresses this point:

“All viruses and lentivectors were titered by infection/transduction of known numbers of relevant target cells with known volumes of viral stocks under standard experimental conditions, followed by flow cytometry for SBP-ΔLNGFR or EGFP plus/minus CD4 at 48 hrs to identify the fraction of infected cells (f) containing at least one transcriptionally active provirus (SBP-ΔLNGFR or EGFP positive plus/minus CD4 low). The number of infectious/transducing units present was then calculated by assuming a Poisson distribution (where f = 1-e^-MOI^). Typically, a dilution series of each viral stock was tested, and titer determined by linear regression of -ln(1-f) on volume of virus.

Unlike CEM-T4s, permissivity of infected primary T cells varies between donors/experiments, and the maximum fraction of infected cells in viral dilution series is often around 50% for single round infections, even at high MOI. In practice, we therefore aimed to use sufficient infectious/transducing units to achieve approximately 30% infection, corresponding to a “nominal” MOI ≤0.5 (assuming a Poisson distribution). This ensured that, even if only 50% of cells were permissive, the “effective” MOI would still be ≤1.”

We have also reviewed every incidence of “multiplicity of infection” or MOI in the manuscript and made some small textual alterations to ensure that the usage is clear, consistent and conservative (for instance, where appropriate, referring to “MOI ≤1” rather than “MOI <1”).

Finally, and as per our responses reviewer #2, we agree that there is the potential for artefacts related to pre-existing proteomic differences in infected (permissive) cells, compared with the mock-infected bulk population. We therefore addressed this point directly with the SBP-ΔLNGFR control proteomic experiment outlined in Figure 3—figure supplement 4 (proteins regulated by transduction with control lentivectors) and Figure 3—source data 1 and described above.

2) A key goal of this study is to provide a resource for examining which proteins are manipulated by HIV infection. With this in mind, could the authors annotate the data in Figure 3C with more detail. Each outlier circle could be numbered and a table of gene names provided. I appreciate that the authors are presenting all of their data and are not hiding anything. But I feel the outliers are what people really want to know about and these could be labeled here. In fact, any opportunity to label the volcano plots would enormously improve the accessibility and thus the likelihood that the field will chase these hits up mechanistically.

We tried to label outliers in the volcano plots as suggested, but over and above the labels currently provided e.g. HIV proteins, ARID5A/PTPN22, we found it to be impossible to do this systematically i.e. not “cherry picking” without the figures becoming cluttered and losing clarity.

Instead, we have therefore provided an additional interactive filter table focusing on the 650 proteins significantly regulated by HIV (Figure 3—source data 1), and colour-coded the proteins as in the volcano plots in Figure 3C and pie chart in Figure 3—figure supplement 3B i.e. green, controls/known accessory protein targets; gold, novel Vpr targets/Vpr-dependent changes (Greenwood et al., 2018); and red, novel/uncharacterised changes. It is easy to identify outliers from this table on the basis of either fold change or q value.

3) It’s not immediately obvious what "pos" and "neg" mean in Figure 2A. For clarity, label "HIV+" and "HIV-" instead and change the text in subsection “Time-dependent proteomic remodelling during HIV infection of primary T cells” to "whole cell lystates from both HIV positive and HIV negative populations using.….

We have changed these labels to “LNGFR+” and “LNGFR-” (for consistency with the remainder of the figure, including flow cytometry data) and added additional information to the key for Figure 3C (defining “LNGFR+ (HIV-infected, selected) and “LNGFR- (uninfected, flow-through)” cells.

4) Provide key for blue and red lines in Figure 2C-E on Figure and in legend.

A key is provided for these lines and described in the Figure legend. We have adjusted slightly to enhance clarity.

5) Subsection “Design and construction of the HIV-AFMACS reporter virus” submit the sequence of the construct to GenBank and provide accession number which is more useful than having the seq in a Figure.

As above, we have submitted the full sequence to GenBank, and provided the accession number in the text (MK435310).

6) Subsection “Proteins and pathways regulated by HIV in primary T cells from multiple donors”, its useful to label the x for the constructs that didn't work. Knowing what didn't work is useful, particularly for those that were completely dead vs a bit defective. Figure 1—figure supplement 1B.

All constructs have been labelled as requested (Figure 1—figure supplement 1A-B).

7) Can the authors comment on the impact of T cell activation of HIV permissivity. It’s not totally clear why T cells have to be activated to make them permissive for HIV infection. SAMHD1 has a lot to do with it and the authors make this point. In these experiments T cell receptor is crosslinked with anti-CD3, anti-CD28 dyna beads. This presumably doesn't happen in vivo, yet T cells are permissive. I imagine the authors have thought about this and I would be interested to hear their thoughts, perhaps in the Discussion section. Is there value in their data comparing unactivated and activated cells to consider what's driving permissivity? How do their data compare with any published literature on in vivo activated T cells, indeed is there any data on this? I'm interested to hear what they think and whether they think their data can be used to illuminate the changes that occur on TCR cross linking that drive permissivity. Would a figure considering this point be valuable?

We agree with the reviewer that this is an interesting area. Determinants of HIV permissivity are clearly not limited to SAMHD1, and may be underpinned by some of the proteomic changes described in this study e.g. associated with SAMHD1-independent metabolic reprogramming (Taylor et al., 2015). Unfortunately, it is difficult to identify individual, novel candidates from this sort of dataset, because (as alluded to by the reviewer) *in vitro* T cell activation typically results in very extensive proteomic remodelling. We have added a figure (Figure 2—figure supplement 2) and additional text to illustrate this point (subsection “Time-dependent proteomic remodelling during HIV infection of primary T cells”):

“In total, we quantitated 9070 cellular proteins across 10 different conditions. As previously reported (Geiger et al., 2016), T cell activation itself caused extensive proteomic remodelling, with relative abundances of 2677/9070 (29%) proteins changing by > 2-fold in activated vs resting cells (Figure 2—figure supplement 2).”

Rather like the reviewer’s suggestion, a very recent paper did seek to correlate transcriptional and metabolic differences in *ex vivo* CD4^+^ T cell subsets with/without sub-maximal *in vitro* activation with differences in susceptibility to HIV infection, again identifying a dependency on levels of metabolic activity (Valle-Casuso et al., 2018). We are not aware of any conceptually similar, high coverage proteomic datasets for comparison with our own. In fact, notwithstanding HIV, we believe that the results presented in this study comprise the most accessible high coverage proteomic atlas of resting vs activated primary human CD4^+^ T cells currently available and may certainly be used as such.